# Random Policy Evaluation Uncovers Policies of Generative Flow Networks

**Haoran He** [1]  **Emmanuel Bengio** [2]  **Qingpeng Cai** [3]  **Ling Pan** [1]

## Abstract

The Generative Flow Network (GFlowNet) is a probabilistic framework in which an agent learns a stochastic policy and flow functions to sample objects proportionally to an unnormalized reward function. A number of recent works explored connections between GFlowNets and maximum entropy (MaxEnt) RL, which modifies the standard objective of RL agents by learning an entropy-regularized objective. However, the relationship between GFlowNets and standard RL remains largely unexplored, despite the inherent similarities in their sequential decision-making nature. While GFlowNets can discover diverse solutions through specialized flow-matching objectives, connecting them can simplify their implementation through established RL principles and improve RL's diverse solution discovery capabilities. In this paper, we bridge this gap by revealing a fundamental connection between GFlowNets and one RL's most basic components – policy evaluation. Surprisingly, we find that the value function obtained from evaluating a uniform policy is closely associated with the flow functions in GFlowNets through the lens of flow iteration under certain structural conditions. Building upon these insights, we introduce a rectified random policy evaluation (RPE) algorithm, which achieves the same reward-matching effect as GFlowNets based on simply evaluating a fixed random policy in these cases, offering a new perspective. Empirical results across extensive benchmarks demonstrate that RPE achieves competitive results compared to previous approaches, shedding light on the previously overlooked connection between (non-MaxEnt) RL and GFlowNets.

[1]Hong Kong University of Science and Technology [2]Valence Labs [3]Kuaishou Technology. Correspondence to: Ling Pan <ling-pan@ust.hk>.

*Proceedings of the 42ⁿᵈ International Conference on Machine Learning*, Vancouver, Canada. PMLR 267, 2025. Copyright 2025 by the author(s).

## 1. Introduction

Generative Flow Networks (GFlowNets) (Bengio et al., 2021; 2023) have emerged as a powerful probabilistic framework for object generation and sampling from complex distributions, which can be seen as a variant of amortized variational inference methods (Ganguly et al., 2022). In GFlowNets, an agent learns a stochastic policy $\pi(x)$ and flow functions to sample objects $x \in \mathcal{X}$ proportionally to an unnormalized reward function $R(x)$. GFlowNets are related to Markov chain Monte Carlo (MCMC) methods (Metropolis et al., 1953; Hastings, 1970; Andrieu et al., 2003), but do not rely on Markov chains that make small and local steps, which often leads to inefficient sampling in high-dimensional discrete spaces due to their local exploration nature. Therefore, they can generalize and amortize the cost of sampling without suffering from the mixing problem (Salakhutdinov, 2009; Bengio et al., 2013; 2021).

GFlowNets reformulate the sampling problem as a sequential decision-making process: objects $x \in \mathcal{X}$ are constructed incrementally through a sequence of steps, where at each step the GFlowNets agent adds an element to the current construction. The sequential nature of GFlowNets is closely related to the decision-making processes in reinforcement learning (RL) (Sutton & Barto, 1998), whose training objectives (Bengio et al., 2021) were also motivated by the temporal difference methods. However, GFlowNets pursue a different goal: instead of maximizing rewards as in standard RL, they aim to match the underlying reward distribution ($\pi(x) \propto R(x)$) (Bengio et al., 2021). This enables GFlowNets to discover diverse, high-reward candidates by sampling them proportionally to their rewards (reward-matching), proving particularly valuable in scenarios with uncertain or imperfect rewards, such as drug discovery (Jain et al., 2023a). This capability has led to significant advances in various challenging domains, including molecule generation (Bengio et al., 2021), biological sequence design (Jain et al., 2022; Chen & Mauch, 2023), Bayesian structure learning (Deleu et al., 2022), and combinatorial optimization (Zhang et al., 2024; 2023b).

Recent works have explored the connections between GFlowNets and maximum entropy (MaxEnt) RL (Tiapkin et al., 2024; Deleu et al., 2024; Mohammadpour et al., 2024), a variant of RL that modifies the standard objective

by incorporating an entropy regularization term (Haarnoja et al., 2017b). These works reveal that GFlowNets' reward-matching behavior can emerge from entropy-regularized objectives, providing valuable insights into the relationship between the two frameworks. However, the connection between GFlowNets and standard (non-MaxEnt) RL remains largely unexplored and poorly understood, despite the fact that both are rooted in sequential decision-making. Understanding this relationship can unlock new possibilities and further advance both fields by combining their strengths: enabling GFlowNets to leverage well-established RL techniques for improved sampling efficiency and stability (Lau et al., 2024), while providing new perspectives on exploration and diversity in RL through GFlowNets' reward-matching behavior (Hu et al., 2024).

In this paper, we uncover a new connection that bridges this gap through one of the most basic components of RL: policy evaluation. To establish this connection, we first reformulate GFlowNets training from a dynamic programming perspective through flow iteration, which paves the way for understanding their relationship. While previous works have primarily focused on modifications to RL objectives and introducing additional entropy regularization terms, we show that policy evaluation, which is often viewed as a fundamental building block for estimating the expected value of a given policy, can be naturally connected to GFlowNets. Specifically, we discover that the resulting value function obtained from evaluating a uniform random policy under reward transformation is closely associated with the flow functions in GFlowNets under specific structural conditions. Our findings reveal an unexpected connection and bridge the gap between these two frameworks, offering a more comprehensive understanding of their underlying connections than previously recognized. Building upon this insight, we introduce a rectified random policy evaluation (RPE) algorithm based on simply evaluating a fixed random policy, providing a straightforward implementation path for GFlowNets in these applicable settings while maintaining the same reward-matching capability.

To validate our findings, we conduct extensive experiments and compare RPE with GFlowNets (Bengio et al., 2021; Malkin et al., 2022; Bengio et al., 2023; Madan et al., 2023) and MaxEnt RL (Haarnoja et al., 2017a; Vieillard et al., 2020). Our results demonstrate that RPE achieves competitive performance compared to previous approaches in such domains, highlighting the effectiveness of our proposed method, and also shed light on the previously overlooked yet fundamental connection between RL and GFlowNets.

## 2. Background

### 2.1. Generative Flow Networks (GFlowNets)

Consider a directed acyclic graph (DAG) $\mathcal{G} = \{\mathcal{S}, \mathcal{A}\}$, where $\mathcal{S}$ and $\mathcal{A}$ represent the state and action spaces. The

objective of GFlowNets is to learn a stochastic policy $\pi$ that constructs discrete objects $x \in \mathcal{X}$ with probability proportional to the reward function: $R$, i.e., $\pi(x) \propto R(x)$. The agent generates objects through a sequential process, and adds a new element to the current state at each timestep $t$. The sequence of states transitions from the initial state to a terminal state is referred to as a trajectory, denoted by $\tau = (s_0 \to ... \to s_n)$, where $\tau \in \mathcal{T}$ belongs to the set of all possible trajectories $\mathcal{T}$. Bengio et al. (2021) introduce the definition of the trajectory flow, represented by the function $F : \mathcal{T} \to \mathbb{R}_{\geq 0}$, which assigns a non-negative real value to each trajectory. The state flow, denoted by $F(s)$, is defined as the sum of flows of all trajectories passing through state $s$, i.e., $F(s) = \sum_{\tau \ni s} F(\tau)$. The edge flow $F(s \to s')$ is the sum of flows of all trajectories containing the transition from state $s$ to state $s'$, which is defined as $F(s \to s') = \sum_{\tau \ni s \to s'} F(\tau)$. We can then define the forward policy $P_F(s'|s) = F(s \to s')/F(s)$, which determines the transition probabilities from a state $s$ to its possible children states $s'$. In addition, we define the backward policy $P_B(s|s') = F(s \to s')/F(s')$, which specifies the likelihood of reaching the parent state $s$ from the current state $s'$. A flow is considered consistent if the total incoming flow for a state matches the total outgoing flow for all internal states $s$, i.e.,

$$\sum_{s'' \to s} F(s'' \to s) = F(s) = \sum_{s \to s'} F(s \to s'). \quad (1)$$

It is proven in (Bengio et al., 2021; 2023) that for consistent flows, the policy can sample objects $x$ with probability proportional to $R(x)$ and therefore match the underlying reward distribution.

**Flow Matching.** Flow matching (FM) (Bengio et al., 2021) parameterizes the edge flow function by $F_\theta(s, s')$, with $\theta$ denoting the learnable parameters, and aims to optimize $F_\theta(s, s')$ for satisfying the flow consistency constraint. The FM loss is defined as $\mathcal{L}_{\text{FM}}(s) = (\log \sum_{s'' \to s} F_\theta(s'', s) - \log \sum_{s \to s'} F_\theta(s, s'))^2$ for non-terminal states, which is the squared difference between the sum of incoming flows and the sum of outgoing flows (optimized in the log-scale due to stability issues). The term $\sum_{s \to s'} F_\theta(s, s')$ is replaced by $R(s)$ if $s$ is a terminal state.

**Detailed Balance.** Detailed balance (DB) parameterizes a state flow model $F_\theta$, a forward policy model $P_{F_\theta}$, and a backward policy model $P_{B_\theta}$ (Bengio et al., 2023), which aims to minimize the loss defined as $\mathcal{L}_{\text{DB}}(s, s') = (\log(F_\theta(s) P_{F_\theta}(s'|s)) - \log(F_\theta(s') P_{B_\theta}(s|s')))^2$, considering the flow consistency constraint at the edge level, and also guarantees correct sampling from the target distribution.

**(Sub) Trajectory Balance.** Malkin et al. (2022) propose a trajectory-level optimization which is analogous to the Monte Carlo approach (Hastings, 1970) in RL, defined as $\mathcal{L}_{\text{TB}}(\tau) = (\log Z_\theta \prod_{t=0}^{n-1} P_{F_\theta}(s_{t+1}|s_t)) - $

$\log R(x) \prod_{t=0}^{n-1} P_{B_\theta}(s_t|s_{t+1}))^2$, that involves training the total flow $Z$, the forward and backward policies. To mitigate the large variance, SubTB (Madan et al., 2023) optimizes the flow consistency constraint in sub-trajectory levels. Specifically, it considers all possible $O(n^2)$ sub-trajectories $\tau_{i:j} = \{s_i, \cdots, s_j\}$, and obtain the objective defined as $\mathcal{L}_{\text{SubTB}}(\tau) = \sum_{\tau_{i:j} \in \tau} w_{ij} \left( \log \frac{F(s_i) \prod_{t=i}^{j-1} P_F(s_{t+1}|s_t)}{F(s_j) \prod_{t=i}^{j-1} P_B(s_t|s_{t+1})} \right)^2$, where $w_{ij}$ represents the weight for $\tau_{i:j}$.

## 2.2. Reinforcement Learning (RL)

A Markov decision process (MDP) is defined as a 5-tuple $(\mathcal{S}, \mathcal{A}, P, r, \gamma)$, where $\mathcal{S}$ represents the set of states, $\mathcal{A}$ represents the set of actions, $P : \mathcal{S} \times \mathcal{A} \to \mathcal{S}$ denotes the transition dynamics, $r$ is the reward function, and $\gamma$ is the discount factor. In an MDP, the RL agent interacts with the environment by following a policy $\pi$, which maps states to actions. The value function in a state $s$ for a policy $\pi$ is defined as the expected discounted cumulative reward the agent receives starting from the state $s$, i.e., $V^\pi(s) = \mathbb{E}_\pi[\sum_{t=0}^{\infty} \gamma^t r(s_t, a_t)|s_0 = s]$. The goal of RL is to find an optimal policy that maximizes the value function at all states. We consider the RL setting consistent with GFlowNets (as in Tiapkin et al. (2024)), with deterministic transitions and the discount factor to be 1, and without intermediate rewards as in Bengio et al. (2013). It is worth noting that GFlowNets can also be extended to stochastic tasks (Pan et al., 2023b; Zhang et al., 2023a), albeit we focus on the more standard deterministic setting in this work. In GFlowNets, the reward is obtained at the terminal state, while the reward typically occurs at transitions in RL. To bridge this gap, we define the value of terminal states $V(x)$ as $R(x)$.

Policy evaluation (Sutton & Barto, 1998) in the dynamic programming literature considers how to compute the value function for an arbitrary policy $\pi$, which is also referred to as the prediction problem. The iterative policy evaluation algorithm is summarized in Algorithm 1.

---

**Algorithm 1** Policy Evaluation

---

**input** The policy $\pi$ to be evaluated; a small threshold $\theta$ for the accuracy of estimation
1: Initialize value functions $V(s)$ arbitrarily for $s \in \mathcal{S}$, and $V(x) = R(x)$ for $x \in \mathcal{X}$
2: **repeat**
3:     $\Delta \leftarrow 0$
4:     **for** $s \in \mathcal{S} \setminus \mathcal{X}$ **do**
5:         $v \leftarrow V(s)$
6:         $V(s) \leftarrow \sum_a \pi(a|s)(r + \gamma V(s'))$
7:         $\Delta \leftarrow \max(\Delta, |v - V(s)|)$
8:     **end for**
9: **until** $\Delta < \theta$

---

**Maximum entropy RL.** Maximum entropy RL (Neu et al.,

2017; Haarnoja et al., 2017a; Geist et al., 2019) considers an entropy-regularized objective augmented by the Shannon entropy, i.e.,

$$V_{\text{Soft}}^\pi(s) = \mathbb{E}_\pi \left[ \sum_{t=0}^{\infty} \gamma^t r(s_t, a_t) + \lambda \mathcal{H}(\pi(s_t))|s_t = s \right], \tag{2}$$

where $\lambda$ is the coefficient for entropy regularization. Schulman et al. (2017a) show that it corresponds to $Q_{\text{Soft}}^*(s, a) = r(s, a) + \gamma \mathbb{E}_{s' \sim P(s,a)}[\lambda \log(\sum_{a'} \exp(Q_{\text{Soft}*}(s', a')/\lambda))]$, with the Boltzmann softmax policy $\pi_{\text{Soft}}^*(a|s) = \exp(\frac{1}{\lambda} Q_{\text{Soft}}^*(s, a) - V_{\text{Soft}}^*(s))$. Littman (1996) introduces the generalized Q-function, which considers using a generalized operator $\otimes$ for updating the Q-values, i.e., $Q(s, a) \leftarrow r(s, a) + \gamma \sum_{s' \in \mathcal{S}} P(s'|s, a) \otimes_{a'} Q(s', a')$. Song et al. (2019); Pan et al. (2020b;a; 2021) study the Boltzmann softmax operator, defined as $\otimes_a Q(s, a) = \frac{\sum_a \exp(\beta Q(s,a))Q(s,a)}{\sum_a \exp \beta Q(s,a)}$, where $\beta$ denotes the temperature (usually set with a non-zero value). When $\beta$ approaches $\infty$, it corresponds to the max operator as typically used in standard Q-learning (Mnih et al., 2013). On the other hand, when $\beta$ approaches 0, it corresponds to the mean or average operator.

## 3. Related Work

**Generative Flow Networks (GFlowNets).** Bengio et al. (2021) introduce GFlowNets as a framework for learning stochastic policies that generate objects $x$ through a sequence of decision-making steps, aiming to sample $x$ with probability proportional to the reward function. GFlowNets have demonstrated remarkable success in various domains, including molecule generation (Bengio et al., 2021), biological sequence design (Jain et al., 2022), Bayesian structure learning (Deleu et al., 2022), combinatorial optimization (Zhang et al., 2023b; 2024), and fine-tuning language models (Li et al., 2023; Hu et al., 2024; Lee et al., 2024; Yun et al., 2025), showcasing their potential for discovering high-quality and diverse solutions. Recent research has focused on providing theoretical understandings of GFlowNets by exploring their connections to variational inference (Zimmermann et al., 2022; Malkin et al., 2023; Niu et al., 2024), generative models (Zhang et al., 2022), and Markov chain Monte Carlo methods (Deleu & Bengio, 2023). Additionally, since the introduction of the flow matching learning objective (Bengio et al., 2021), efforts have been made to enhance the learning efficiency of GFlowNets, tackle large variance (Malkin et al., 2022; Bengio et al., 2023; Madan et al., 2023), improve exploration (Pan et al., 2022; Lau et al., 2024), enable more efficient credit assignment (Pan et al., 2023a; Jang et al., 2024), and extend to stochastic practical environments (Pan et al., 2023b; Zhang et al., 2023a), largely motivated by the development in the RL literature. Temporal-difference methods in reinforcement

learning (RL) (Sutton, 1988) serve as a significant inspiration for GFlowNets (Bengio et al., 2023). There have been a number of recent works drawing connections between GFlowNets and maximum entropy (MaxEnt) RL (Tiapkin et al., 2024; Mohammadpour et al., 2024; Deleu et al., 2024), but they are limited to considering an entropy-regularized objective that differs from the goal of standard RL. This work establishes a link between GFlowNets and standard (non-MaxEnt) RL through one of its most basic building blocks of policy evaluation for a random policy.

**Reinforcement Learning (RL).** In RL, the problem is typically formulated as a Markov decision process (MDP) with states and actions defined similarly to the directed acyclic graph representation in GFlowNets. The agent learns a deterministic optimal policy to maximize the cumulative return (Sutton & Barto, 1998). Maximum-entropy (MaxEnt) RL (Haarnoja et al., 2017a), also known as soft RL or entropy-regularized RL, optimizes an entropy-regularized objective (Fox et al., 2015; Haarnoja et al., 2017b), where the agent seeks to maximize both the reward and action entropy, which falls under the broader domain of regularized MDPs (Neu et al., 2017; Geist et al., 2019). Soft Q-learning (Haarnoja et al., 2017b) is a popular instance of MaxEnt RL, which employs a log-sum-exp operator instead of the max operator commonly used in Q-learning (Mnih et al., 2013), along with a Boltzmann softmax policy (Schulman et al., 2017a; Pan et al., 2020a). Related studies have investigated alternative operators for learning the value function, demonstrating that the Boltzmann softmax operator (Song et al., 2019; Pan et al., 2020a) can mitigate the estimation bias (Pan et al., 2020a; 2021) in popular RL algorithms, when using a non-zero temperature parameter. Recently, Laidlaw et al. (2023) have shown that acting greedily with respect to the value function for a uniform policy can be as competitive as proximal policy optimization (PPO) (Schulman et al., 2017b) in several standard game environments, which highlights the potential of simple, uninformed learning strategies to achieve strong performance.

## 4. Rectified Random Policy Evaluation

There have been a number of recent works (Tiapkin et al., 2024; Deleu et al., 2024) exploring the connections of GFlowNets (Bengio et al., 2023) and maximum entropy (MaxEnt) or soft RL (Haarnoja et al., 2017b; 2018; Geist et al., 2019)), a variant of RL that modifies the standard objective with entropy regularization. Specifically, Tiapkin et al. (2024) show that GFlowNets can be viewed as MaxEnt RL with a particular intermediate reward correction $r(s \rightarrow s') = \log P_B(s|s')$, where the soft value function $V_{\text{soft}}(s)$ corresponds to the logarithm of the state flows $F(s)$ in GFlowNets, i.e., $V_{\text{soft}}(s) = \log F(s)$. Despite these findings, the connection between GFlowNets and standard (non-MaxEnt) RL remains largely unexplored, despite both

frameworks being rooted in temporal difference learning and sequential decision-making.

In this section, we establish a surprisingly simple yet fundamental connection between GFlowNets and standard RL by returning to one of its most basic building blocks: policy evaluation. We present a novel connection between GFlowNets and policy evaluation under random policies, by analyzing the theoretical equivalence under certain structural conditions between flow functions through flow iteration and (scaled) value functions obtained from evaluating a fixed random policy in Section 4.1, which holds unconditionally for tree structures and extends to non-tree DAGs considering uniform backward policies under path-invariance conditions. Building upon these insights, we introduce a rectified random policy evaluation method that provides a simple equivalent alternative to existing GFlowNets training methods in cases where the equivalence holds while achieving the same reward-matching effect as GFlowNets in Section 4.2.

### 4.1. Connections

To bridge the gap between GFlowNets and RL, we first establish GFlowNets training from a dynamic programming (DP) (Barto, 1995) perspective, which paves the way for understanding their relationship as DP principles form the foundation of most RL algorithms (Sutton & Barto, 1998).

---
**Algorithm 2** Flow Iteration
***

**input** The backward policy $P_B$; a small threshold $\theta$ for estimation accuracy

1: Initialize flow functions $F(s)$ arbitrarily for $s \in \mathcal{S}$, and $F(x) = R(x)$ for $x \in \mathcal{X}$
2: **repeat**
3:     $\Delta \leftarrow 0$
4:     **for** $s \in \mathcal{S} \setminus \mathcal{X}$ **do**
5:         $f \leftarrow F(s)$
6:         $F(s) \leftarrow \sum_{s'} P_B(s|s')F(s')$
7:         $\Delta \leftarrow \max(\Delta, |f - F(s)|)$
8:     **end for**
9: **until** $\Delta < \theta$

---

To formalize the DP perspective of GFlowNets, we introduce the Flow Iteration algorithm as outlined in Algorithm 2. Specifically, Flow Iteration estimates the state flow $F(s)$ for non-terminal states $s$ based on its possible children states $s'$ and the backward policy, which is defined as $F(s) = \sum_{s'} P_B(s'|s)F(s')$ (following the flow consistency principle in the state-edge level). This formulation shares certain computational characteristics with policy evaluation in RL: Flow iteration propagates flow values from child to parent states using the backward policy $P_B$, while policy evaluation propagates values from successor to current states using the (forward) policy $\pi$. Note that we consider uniform backward policies for flow iteration in

non-tree DAGs in this work.

Based on this, we now investigate the relationship between flow functions and value functions considering the structural similarities between flow iteration and policy evaluation. We systematically investigate this relationship, beginning with the simpler case of tree-structured graphs (Bengio et al., 2021) and then exploring the more challenging non-tree-structured directed acyclic graph (DAG) cases.

### 4.1.1. TREE DAG

We begin our analysis with tree-structured DAGs (Hu et al., 2024) that serve as a foundation for analyzing more general DAGs. Our key insight is that GFlowNets' flow function $F(s)$ and value functions $V(s)$ in RL share an interesting connection when considering the simplest possible policy - a uniform random policy, and a principled reward transformation considering a scaling factor of the number of available actions in different states. This connection emerges naturally from the structural similarities between our flow iteration and policy evaluation. Specifically, if we scale the original reward function $R(x)$ for terminal states $x$ in flow iteration by the product of available actions along the path to $x$, which accounts for the branching structure of the decision process, we observe an equivalence between $F(s)$ and $V(s)$. This relationship not only provides a new interpretation of GFlowNets but also reveals how standard RL can be repurposed to achieve the same reward-matching behavior of GFlowNets. In Theorem 4.1, we restate and generalize the relation between $V(s)$ (obtained by policy evaluation for a uniform policy) and $F(s)$ (obtained by our proposed flow iteration), extending initial observations made by Bengio et al. (2021), where the proof can be found in Appendix A.

**Theorem 4.1** (Generalization of Bengio et al. (2021))**.** *Let $A(s)$ denote the number of available actions at state $s$, and $R(x)$ be the reward function for terminal states. Let $F(s_t)$ be the state flow function obtained from GFlowNet's flow iteration that samples proportionally from $R(x)$, and $V(s_t)$ be the value function under a uniform policy with transformed rewards $R'(x) = R(x) \prod_{i=0}^{t-1} A(s_i)$. Then, for all $s_t$, $V(s_t) = F(s_t) \prod_{i=0}^{t-1} A(s_i)$.*

**Remark.** This reveals an interesting connection: GFlowNets' sophisticated reward-matching capability can be achieved through a basic operation in RL – policy evaluation of a fixed uniform policy with an appropriately transformed reward structure. Flow iteration, when viewed through the lens of policy evaluation, establishes a bridge between GFlowNets and standard RL. This provides a different perspective from a number of previous works connecting GFlowNets to MaxEnt RL (Tiapkin et al., 2024; Deleu et al., 2024), which incorporates entropy regularization into the standard RL objective. Our finding suggests that the flow-matching objective in GFlowNets can be reinterpreted as a

specific form of value estimation in RL with a specific transformation. It also broadens the understanding of what can be achieved with policy evaluation only, unlike typical RL that requires iterative policy evaluation and policy improvement (policy iteration) for reward maximization (Sutton, 1988).

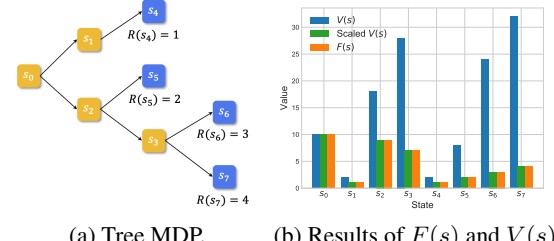

(a) Tree MDP.          (b) Results of $F(s)$ and $V(s)$.

Figure 1: Comparison of random policy evaluation and flow iteration in a tree-structured DAG example.

**Empirical Validation.** Consider a tree-structured DAG as shown in Figure 1(a), where terminal states $x$ (blue squares) are associated with rewards $R(x)$. To validate our theoretical findings, we compare the flow function $F(s)$ obtained through flow iteration using the original reward $R(x)$ with the value function $V(s)$ computed through policy evaluation of a uniform policy using transformed rewards $R'(x)$. The resulting flows $F(s)$ and values $V(s)$ for each state are illustrated in Figure 2(a) and Figure 2(c), respectively. As summarized in Figure 1(b), both $F(s_0)$ and $V(s_0)$ yield identical values, correctly estimating the total flow. For all other states, $F(s)$ exactly matches $V(s)$ after accounting for the scaling factor $\prod_{i=0}^{t-1} A(s_i)$. This empirically confirms that standard policy evaluation for a simple random policy, when configured with our transformed reward structure, learns the same flow functions for achieving the reward-matching capability as GFlowNets. Our approach differs from (Tiapkin et al., 2024) in that we do not consider $\log P_B$ as intermediate rewards and operate in the log-scale, where we instead directly use a transformed terminal reward and a scaling factor computed when we collect the trajectory. This can be illustrated in Figure 2(b), which shows the result of the MaxEnt RL formulation (Tiapkin et al., 2024) that requires intermediate reward corrections ($r = 0$ here due to the tree structure, which can be non-zero in non-tree structured DAG cases) at each transition and operates in log-scale.

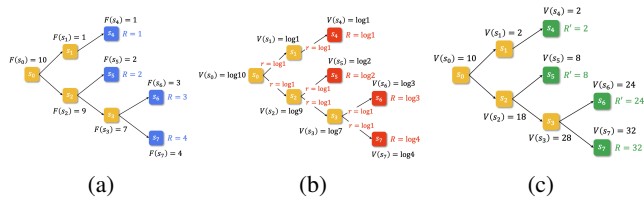

(a)                    (b)                    (c)

Figure 2: Illustration of flow and value for tree cases. (a) Flow iteration. (b) MaxEnt RL with intermediate reward correction in log-scale. (c) Random policy evaluation with transformed rewards.

### 4.1.2. NON-TREE DAG

Building upon the insights gained from the simpler tree-structured case, we now extend our analysis to the more general and challenging setting of non-tree-structured DAGs, which is a natural setting for GFlowNets applications where states can have multiple parent states.

In Theorem 4.2, we present a connection between GFlowNets' flow iteration procedure with uniform backward policy and random policy evaluation in the context of general non-tree DAGs, and establish conditions under which the equivalence holds. The proof can be found in Appendix A.

**Theorem 4.2.** *Let $A(s)$ and $B(s)$ denote the number of outgoing and incoming actions at state $s$ respectively. For any trajectory $\tau$ visiting state $s_t$, its branching ratio up to $s_t$ is defined as $g(\tau, s_t) = \prod_{i=0}^{t-1} \frac{A(s_i)}{B(s_{i+1})}$. Let $F(s_t)$ be the state flow function obtained from GFlowNet's flow iteration with uniform backward policy that samples proportionally from the reward function $R(x)$, and $V(s_t)$ be the value function under a uniform policy with transformed rewards $R'(x) = R(x)g(\tau, x)$. If any trajectories $\tau_1$ and $\tau_2$ that visits $s_t$ satisfy $g(\tau_1, s_t) = g(\tau_2, s_t)$, then for all $s_t$, $V(s_t) = F(s_t)g(\tau, s_t)$[1].*

**Remark.** Theorem 4.2 establishes the relationship between GFlowNets' flow iteration with uniform backward policy and random policy evaluation to non-tree DAG cases under a path-invariance condition, which reveals a consistent pattern: the state flow function $F(s)$ can be expressed as a scaled version of the state value function $V(s)$ under a uniform policy with a transformed reward function. The scaling factor $g(\tau, s_t)$ accounts for both outgoing and incoming actions at each state along the trajectory. Different from the tree case which imposes no constraints (due to its unique path property), the non-tree case requires the path-invariance condition, i.e., $g(\tau_1, s_t) = g(\tau_2, s_t)$, which means that any trajectories $\tau_1$ and $\tau_2$ passing through state $s_t$ should yield identical $g(\tau, s)$ values. This ensures that the scaling factor remains consistent across different paths to the same state, which is essential for establishing the equivalence.

This assumption naturally holds in several practical GFlowNets benchmark environments, e.g., the set generation task studied by Pan et al. (2023a) (where the number of parent or children states remains independent of the state and trajectory itself). While this equivalence applies to this meaningful class of DAG structures, it does not extend universally to all possible DAGs, such as the HyperGrid (Bengio et al., 2021) environment, where its boundary effects create path-dependent $g$-values for the same state. In such cases, it prevents the establishment of a direct relationship of GFlowNets with policy evaluation and limits universal applicability, and our analysis characterizes the conditions under which these two frameworks align. We view this limitation as a natural consequence of the simplicity of the method, which is based on simple policy evaluation for a fixed uniform policy. By identifying the structural conditions necessary for equivalence, our work deepens the understanding of both frameworks and reveals a meaningful class of problems where simplified training strategies become feasible and offers new insights.

**Empirical Validation.** To validate this equivalence, we consider the set generation task studied by Pan et al. (2023a). In this task, the agent sequentially generates a set with size $|S|$ from $|U|$ elements. At each timestep, the agent selects an element from $U$ and adds it to the current set (without replacement). The agent receives the reward for constructing the set with exactly $|S|$ elements.

Figure 3 presents the results in the tabular case, where values computed by random policy evaluation or flows computed by flow iteration with uniform backward policies are represented in tables as the state and action spaces are enumerable. This tabular representation eliminates the influence of neural network approximation and sampling errors, providing a clear comparison between the state flow function and the scaled state value function. In Figure 3, the x-axis corresponds to different states, topologically sorted, and the y-axis corresponds to flows $F$ or values $V$. We compare the resulting state flow function $F(s)$, the value function $V(s)$ under the original rewards $R(x)$ from random policy evaluation, and the value function under transformed rewards $R'(x)$ also from random policy evaluation. We observe that the scaled value function, denoted as scaled $V(s)$, aligns perfectly with the state flow function $F(s)$, validating the equivalence.

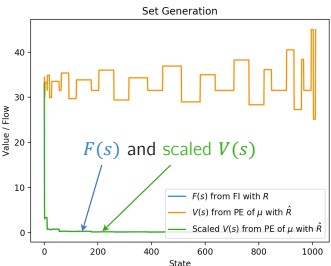

Figure 3: Comparison of random policy evaluation and flow iteration with uniform backward policies in the set generation task.

### 4.2. Rectified Random Policy Evaluation

Based on the above theoretical analysis, we leverage this interesting insight from the connections between flows and values to develop Rectified Random Policy Evalua-

---

[1]Under the path-invariance condition, $g(\tau, s_t)$ is identical for all trajectories $\tau$ reaching $s_t$, and the right-hand side is therefore state-dependent only.

tion (RPE). By rectifying policy evaluation for a uniform policy, RPE achieves the same reward-matching capabilities as GFlowNets that sample proportionally to the rewards ($\pi(x) \propto R(x)$) and discover diverse candidates, while maintaining the simplicity of standard policy evaluation.

The key insight of RPE stems from our theoretical equivalence: by reparameterizing the value function as $V(s) = F(s)g(s)$ and scaling terminal rewards by the $g$-function, we can transform GFlowNet's sophisticated flow-matching objective into a simpler policy evaluation task with transformed rewards $R'(x)$ for a fixed random policy. This transformation is practical as the $g$-function is readily available in standard benchmarks. It is simply the number of available actions at each state along the path in tree-structured problems, while in non-tree DAGs, it is the ratio of outgoing to incoming actions along the path, both of which are predefined by the problem's action space specification. To enable sampling from the learned distribution, we can obtain the forward policy $P_F(s'|s) = \frac{F(s')P_B(s'|s)}{F(s)}$ (Bengio et al., 2023) that inherits GFlowNet's reward-matching properties with a uniform backward policy $P_B$. The complete procedure is detailed in Algorithm 3.

---

**Algorithm 3** Rectified Policy Evaluation

---

**input** Flows $F_\theta(s)$ parameterized by $\theta$, uniform policy $\pi$
**output** Sampling policy $P_F(s'|s) = \frac{F_\theta(s')P_B(s|s')}{F_\theta(s)}$ (with a uniform $P_B$)
1: **for** $t = \{1, \cdots, T\}$ **do**
2:     Sample a trajectory $\tau = \{s_0, \cdots, s_n\}$ using $\pi$
3:     Calculate $g(s)$ for states $s \in \tau$
4:     **for** $s \in \tau$ **do**
5:        **if** $s$ is not a terminal state **then**
6:           $V(s) \leftarrow \sum_a \pi(a|s)F_\theta(s')g(s')$
7:        **else**
8:           $V(s) \leftarrow g(s)R(s)$
9:        **end if**
10:     **end for**
11:     $\theta \leftarrow \arg\min \mathcal{L}_{\text{MSE}}(g(s)F_\theta(s), V(s))$ by an Adam optimizer
12: **end for**

---

**Discussion.** RPE reformulates the GFlowNets training into a random policy evaluation process with rectification, while maintaining equivalent reward-matching capabilities through our established flow-value connections. In RPE, the policy to be evaluated is a fixed uniform policy $\pi$, in contrast to standard GFlowNets and MaxEnt RL that requires estimating flows/values for continuously evolving policies during training, leading to more significant non-stationarity challenge in the learning process (Van Hasselt et al., 2018; Laidlaw et al., 2023). In addition, RPE adopts a simplified parameterization that learns only the flow function $F_\theta$, from which the sampling policy can be directly derived,

which can reduce the potential approximation error from function approximators (Shen et al., 2023). As discussed in Section 4.1.2, the equivalence holds unconditionally in tree-structured problems, making RPE universally applicable to the tree cases. For non-tree DAGs, this equivalence is guaranteed only when the path-invariance property is satisfied, which holds in tasks where state transitions have similar structural properties (e.g., set generation). While this assumption encompasses a broad range of practical applications, investigating scenarios where it may not hold presents a promising direction for future research building upon our analysis. This reformulation reveals previously overlooked connections between GFlowNets and policy evaluation, offering novel perspectives that bridge this gap and advance our understanding of both frameworks.

# 5. Experiments

## 5.1. Experimental Setup

**Baselines.** We extensively compare RPE with GFlowNets with different learning objectives including Flow Matching (FM) (Bengio et al., 2021), Detailed Balance (DB) (Bengio et al., 2023), Trajectory Balance (TB) (Malkin et al., 2022), and Sub-Trajectory Balance (SubTB) (Madan et al., 2023) as introduced in Section 2.1. We use the learned backward policies for GFlowNets baselines (DB, TB, and SubTB) as the default setting, with results using fixed uniform backward policies provided in Appendix C. Additionally, we compare RPE with the maximum entropy (MaxEnt) RL algorithms, i.e., soft DQN (Haarnoja et al., 2017b) and Munchausen DQN (M-DQN;(Vieillard et al., 2020; Tiapkin et al., 2024)), as described in Section 2.2.

**Metrics.** We follow standard evaluation metrics and evaluate each method in terms of the accuracy metric (Shen et al., 2023; Kim et al., 2023), which quantifies how well the learned policy distribution aligns with target reward distribution. Accuracy is calculated by computing the relative error between the sample mean of the reward function $R(x)$ under the learned policy distribution $P_F(x)$ and the expected value of $R(x)$ under the target distribution. The calculation of accuracy is given as $\text{Acc}(P_F(x)) = 100 \times \min\left(\frac{\mathbb{E}_{P_F(x)}[R(x)]}{\mathbb{E}_{p^*(x)}[R(x)]}, 1\right)$, where $p^*(x) = R(x)/Z$ represents the target distribution. We also analyze the number of modes discovered during the course of training (Bengio et al., 2021; Jain et al., 2022), which measures the ability to identify multiple high-reward regions in the solution space.

**Tasks.** We compare RPE against GFlowNets and MaxEnt RL baselines across several GFlowNets tasks, including TF-Bind generation (Shen et al., 2023), RNA design (Kim et al., 2023), and molecule generation (Shen et al., 2023). We consider minimally modified transition dynamics that ensure the path-invariance assumption required in non-tree DAGs

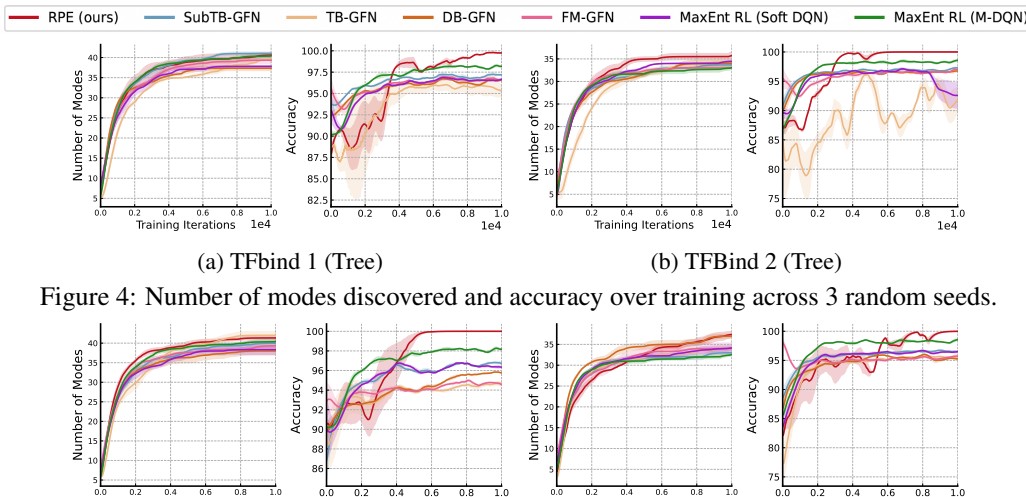

(a) TFbind 1 (Tree)             (b) TFBind 2 (Tree)

Figure 4: Number of modes discovered and accuracy over training across 3 random seeds.

(a) TFBind 1 (DAG)             (b) TFBind 2 (DAG)

Figure 5: Number of modes discovered and accuracy of each method.

can be satisfied for comprehensive evaluation, while maintaining the essential characteristics of these tasks. Specifically, when the sequences contain the same amino acids for adding, we restrict the construction process to only append amino acids. Note that this modification only affects a few border cases, and it maintains the essential characteristics of the original tasks.

**Implementations.** We implement all baselines based on open-source codes from Kim et al. (2023)[2] and Tiapkin et al. (2024)[3]. We run each algorithm with three random seeds and report both the mean and standard deviation of their performance metrics. To ensure a fair comparison, our method and all baselines use the same network architecture, batch size, and other relevant hyperparameters, with a more detailed description in Appendix B due to space limitation.

### 5.2. TF Bind Generation

We first explore the task of generating DNA sequences that exhibit high binding activity with human transcription factors (Jain et al., 2022). The objective of the agent is to discover a diverse set of promising candidates that demonstrate strong binding affinity to the target transcription factor. At each timestep, the agent selects an amino acid $a$ and incorporates it into the currently generated partial sequence. We consider four reward functions from Lorenz et al. (2011). We first study the task of a left-to-right generation of the TF Bind sequence as studied in Malkin et al. (2022), where the agent chooses to append an amino acid $a$ to the end of the current state. This choice of constructive actions leads to a tree-structured problem, as each state only has one

parent state. We then investigate a variant of the prepend-append MDP (PA-MDP) (Shen et al., 2023) for the TF Bind generation task, where we consider a structured version where minimally modified transition dynamics ensure the path-invariance condition as introduced in Section 5.1 (more detailed description can be found in Appendix B). This formulation results in a more complex directed acyclic graph, as opposed to a simple tree, due to the existence of multiple trajectories for each object $x$, which poses significant challenges in the learning process.

Figures 4-5 summarize the results in terms of accuracy and the number of modes discovered during training for each method, considering tree-structured generation and DAG-structured TF Bind generation, respectively. These figures present the results for two reward functions, while the complete results for all four reward functions can be found in Appendix C due to space constraints. As shown, GFlowNets (including FM, DB, TB, SubTB learning objectives), MaxEnt RL (including Soft DQN and Munchausen DQN (Tiapkin et al., 2024)), and our RPE algorithm achieve comparable performance in terms of the number of modes discovered, indicating their ability to effectively capture the multi-modal nature of the reward function, due to their equivalence. In terms of accuracy, we observe that RPE, a simplified learning process of evaluating fixed random policies under appropriate transformation, generally outperforms other baselines by a small margin.

### 5.3. RNA Sequence Generation

In this section, we study a larger practical task of generating RNA sequences. We consider four distinct target transcriptions employing the ViennaRNA package (Lorenz et al., 2011) as studied in Pan et al. (2024), where each task eval-

---

[2] https://github.com/dbsxodud-11/ls_gfn
[3] https://github.com/d-tiapkin/gflownet-rl

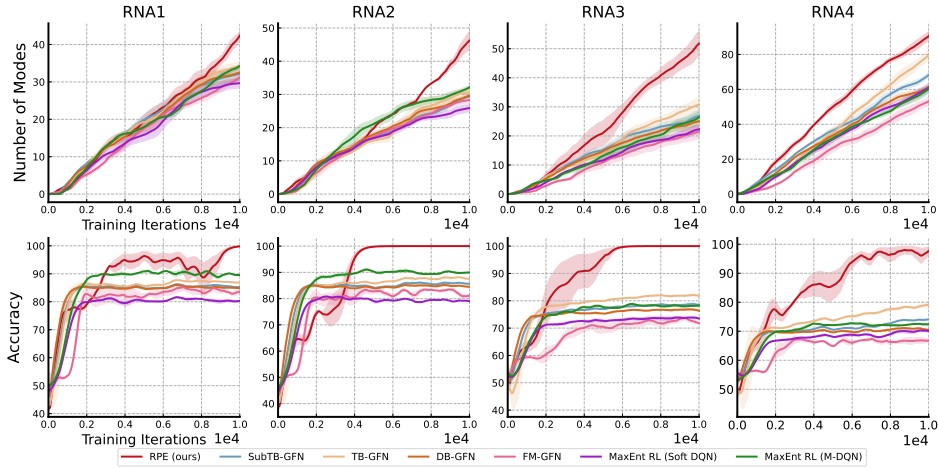

Figure 6: *Top row:* number of modes discovered over training across 3 random seeds. *Bottom row:* Accuracy over training across 3 random seeds.

uates the binding energy with a unique target serving as the reward signal for the agent (Lorenz et al., 2011). We follow the experimental setup as in Section 5.1, and study the variant of prepend-append MDP introduced by Shen et al. (2023). More detailed descriptions of the setup can be found in Appendix B.

The performance of each method in terms of accuracy and the number of modes discovered for each task is shown in Figure 6. The results show that RPE captures the multi-modal reward landscape well, maintaining the key capability of GFlowNets in discovering diverse, high-reward solutions. In addition, in this more challenging task with larger state spaces, we observe that RPE demonstrates stronger performance across both metrics, achieving nearly 100% accuracy while discovering more modes than the baselines, as RPE provides a simpler parameterization for evaluating a fixed random policy different from baseline methods.

### 5.4. Molecule Generation

In this section, we study the task of generating molecule graphs. We study the variant of the QM9 molecule task as studied in prior GFlowNets work (Jain et al., 2023b; Shen et al., 2023; Kim et al., 2023) following the experimental setup as in Section 5.1, where the reward function is defined as the energy gap between the highest occupied molecular orbital and lowest unoccupied orbital (HOMO-LUMO). We employ a pre-trained molecular property prediction model, MXMNet (Zhang et al., 2020), as the reward proxy.

The results are summarized in Figure 7, where RPE consistently maintains the ability to capture multi-modal rewards, demonstrating comparable accuracy to GFlowNet and MaxEnt RL baselines. RPE also discovers modes faster from its fixed policy evaluation framework.

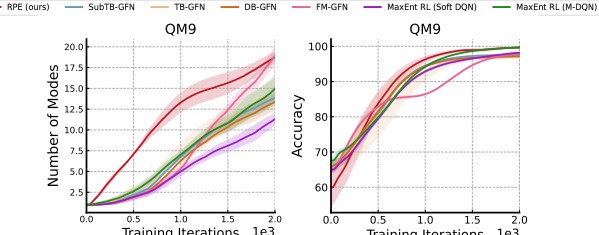

Figure 7: Results in the QM9 molecule generation tasks.

## 6. Conclusion

In this paper, we establish a new connection between GFlowNets and core reinforcement learning concepts through the lens of flow iteration and policy evaluation. Our results reveal that the value function under a uniform policy is intrinsically linked to the flow functions in GFlowNets under certain structural conditions. Based on this insight, we develop Rectified Policy Evaluation (RPE), which reformulates GFlowNets' objectives through a simplified policy evaluation framework. Extensive empirical evaluations on GFlowNets benchmarks demonstrate that RPE is able to capture multi-modal rewards and achieve GFlowNets' sophisticated reward-matching capability through evaluating a fixed uniform policy, providing new insights into understanding both fields.

While there are no structural constraints required for tree-structured environments, the extension to non-tree DAGs requires the path-invariance condition, which may not hold universally across all problem domains. Investigating whether the relationship can be extended to broader classes of environments without restrictive assumptions, exploring connections with non-uniform policies, and developing adaptive methods that can automatically satisfy structural requirements in general settings are interesting future directions.

## Acknowledgment

This work is supported by the National Natural Science Foundation of China 62406266.

## Impact Statement

This paper presents work whose goal is to advance the field of Machine Learning. There are many potential societal consequences of our work, none which we feel must be specifically highlighted here.

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

## A. Proofs of Theorems 4.1-4.2

It can be simply verified that the tree-structured DAG (Theorem 4.1) is a special case of the non-tree-structured DAG (Theorem 4.2), with $B(s_{i+1}) = 1$ and there is only one path from $s_0$ to any states, and $P_B$ is trivial in this case. Therefore, we prove the general non-tree case here with mathematical induction (and the proof for Theorem 4.1 can also be obtained via the following proof since it is a special case).

**Theorem 4.2** *Let $A(s)$ and $B(s)$ denote the number of outgoing and incoming actions at state $s$ respectively. For any trajectory $\tau$ visiting state $s_t$, its branching ratio up to $s_t$ is defined as $g(\tau, s_t) = \prod_{i=0}^{t-1} \frac{|A(s_i)|}{|B(s_{i+1})|}$. Let $F(s_t)$ be the state flow function from GFlowNet's flow iteration with uniform backward policy that samples proportionally from the reward function $R(x)$, and $V(s_t)$ be the value function under a uniform policy with transformed rewards $R'(x) = R(x)g(\tau, x)$. If any trajectories $\tau_1$ and $\tau_2$ that visits $s_t$ satisfy $g(\tau_1, s_t) = g(\tau_2, s_t)$, then for all $s_t$, $V(s_t) = F(s_t)g(\tau, s_t)$.*

*Proof.* For all terminal states $s_n$, by definition, we have that

$$V(s_n) = R'(s_n) = R(s_n)g(\tau, s_n). \tag{3}$$

$$F(s_n) = R(s_n). \tag{4}$$

Thus, $V(s_n) = F(s_n)g(\tau, s_n)$ holds for all terminal states.

Then, for any other state $s_t$, assume all of its children $s_k$ satisfy

$$V(s_k) = F(s_k)g(\tau, s_k). \tag{5}$$

By the definition of the policy evaluation procedure, we have that

$$V(s_t) = \sum_{s_t \to s_k} \frac{V(s_k)}{|A(s_t)|} \tag{6}$$

Combining Eq.(5) and Eq.(6), we get

$$V(s_t) = \sum_{s_t \to s_k} \frac{F(s_k)g(\tau, s_k)}{|A(s_t)|}. \tag{7}$$

By definition, we have that

$$g(\tau, s_k) = \prod_{i=0}^{k-1} \frac{|A(s_i)|}{|B(s_{i+1})|}. \tag{8}$$

Thus, we obtain that

$$\frac{g(\tau, s_k)}{|A(s_t)|} = \frac{g(\tau, s_k)}{|A(s_{k-1})|} = \frac{\prod_{i=0}^{k-2} |A(s_i)|}{\prod_{i=0}^{k-1} |B(s_{i+1})|} = \frac{1}{|B(s_k)|} \prod_{i=0}^{k-2} \frac{|A(s_i)|}{|B(s_{i+1})|}. \tag{9}$$

As

$$g(\tau, s_t) = g(\tau, s_{k-1}) = \prod_{i=0}^{k-2} \frac{|A(s_i)|}{|B(s_{i+1})|}, \tag{10}$$

and combing Eq.(7),(9), and (10), we have that

$$V(s_t) = \sum_{s_t \to s_k} \frac{F(s_k)}{|B(s_k)|} g(\tau, s_t). \tag{11}$$

By the definition of the flow iteration procedure considering uniform backward policies ($P_B(s_t|s_k) = \frac{1}{|B(s_k)|}$), we have that

$$F(s_t) = \sum_{s_t \to s_k} \frac{F(s_k)}{|B(s_k)|}. \tag{12}$$

Combing Eq.(11) and (12), we obtain that

$$V(s_t) = F(s_t)g(\tau, s_t). \tag{13}$$

Therefore, the condition holds for $s_t$. Finally, by induction, $V(s_t) = F(s_t)g(\tau, s_t)$ holds for all states. Note that under the path-invariance condition, $g(\tau, s_t)$ is identical for all trajectories $\tau$ reaching $s_t$, and the right-hand side is therefore state-dependent only.

$\square$

## B. Experimental Setup

### B.1. Implementation Details

- We use an MLP network that consists of 2 hidden layers with 2048 hidden units and ReLU activation (Xu et al., 2015) to estimate flow function $F_\theta$.

- We encode each state into a one-hot encoding vector and feed them into the MLP network.

- We clip gradient norms to a maximum of 10.0 to prevent unstable gradient updates.

- We run all the experiments in this paper with RTX 3090 GPU.

Below we introduce separate details for different benchmarks used in this paper. For thorough comparison across methods considering the path-invariance property, we consider minimally modified transition dynamics that ensure the assumption required for GFlowNets and RL can be satisfied for comprehensive evaluation. Specifically, when the sequences contain the same amino acids for adding, we restrict the construction process to only append amino acids. This modification only affects a few border cases, and it maintains the essential characteristics of the original tasks.

**TF Bind generation.** For the tree-structured TF Bind task, we follow the experimental setup described in Jain et al. (2022); For the graph-structured TF Bind task, we follow the experimental setup described in Shen et al. (2023). We select four tasks defined by different reward functions in Lorenz et al. (2011), i.e., *SIX6_T165A_R1_8mers*, *ARX_P353L_R2_8mers*, *PAX3_Y90H_R1_8mers* and *WT1_REF_R1_8mers*. In this task, we train our model for $1e4$ steps, using the Adam optimizer (Kingma & Ba, 2014) with a $3e-3$ learning rate. We set the reward threshold as 0.8 and the distance threshold as 3 to compute the number of modes discovered during training.

**RNA Sequence generation.** We consider the PA-MDP to generate strings of 14 nucleobases. Following Pan et al. (2024), we present four different tasks characterized by different reward functions, i.e., *RNA1*, *RNA2*, *RNA3* and *RNA4*. In this task, we train our model for $1e4$ steps, using the Adam optimizer (Kingma & Ba, 2014) with a $3e-3$ learning rate. We set the reward exponent as 3. We set the reward threshold as 0.8 and the distance threshold as 3 to compute the number of modes discovered during training. We normalize the reward into $[0.001, 10]$ during training.

**Molecule generation.** The goal of QM9 is to generate a molecule with 5 blocks from 12 building blocks with 2 stems. Following the experimental setup described in Kim et al. (2023), we set the reward exponent as 5. We train our model for $2e3$ steps, using the Adam optimizer (Kingma & Ba, 2014) with a $1e-3$ learning rate. To compute the number of modes discovered during training, we set the reward threshold as 1.4 and the distance threshold as 3. We normalize the reward into $[0.001, 1000]$. This normalization is beneficial for flow regression.

**Baselines.** We implement the MaxEnt RL baselines, including soft DQN and Munchausen DQN, by borrowing the code from `https://github.com/d-tiapkin/gflownet-rl` (Tiapkin et al., 2024). We implement the baselines of GFlowNets based on the codes from `https://github.com/dbsxodud-11/ls_gfn` (Kim et al., 2023).

## C. Additional Experimental Results

**TF Bind** We provide the other two tasks of TF Bind generation benchmark in Fig 8. We find that RPE performs competitively in both tree and graph TFBind tasks.

**RNA generation.** Table 1 and Table 2 summarize the results of the final model in the RNA generation tasks, which clearly show the superior performance of our method RPE in terms of both accuracy and number of modes discovered.

**Baselines with uniform $P_B$.** Furthermore, we conduct a comparison between RPE and GFlowNets methods using a uniform $P_B$. As depicted in Fig. 9, the $P_B$ values in TB-GFN, SubTB-GFN, and DB-GFN are consistently fixed to be uniform. Our observations indicate that while these GFlowNets methods with uniform $P_B$ exhibit rapid convergence, they generally yield inferior performance compared to instances where $P_B$ is learned.

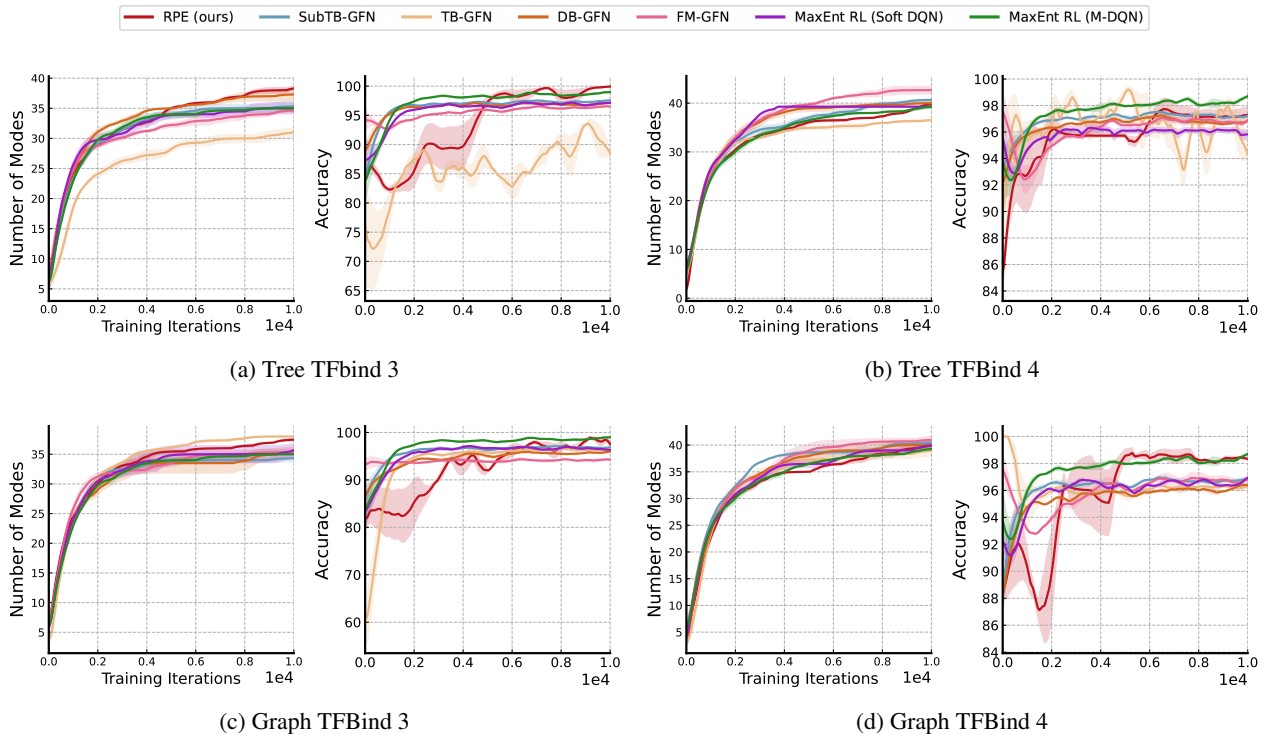

Figure 8: Number of modes discovered and accuracy over training across 3 random seeds.

Table 1: Accuracy in different RNA generation tasks.

|  | L14_RNA1 | L14_RNA2 | L14_RNA3 | L14_RNA4 |
|---|---|---|---|---|
| FM-GFN | $83.54 \pm 1.46$ | $81.16 \pm 1.03$ | $71.82 \pm 0.57$ | $66.69 \pm 1.13$ |
| DB-GFN | $88.48 \pm 0.41$ | $88.49 \pm 0.33$ | $76.35 \pm 0.44$ | $70.53 \pm 0.12$ |
| TB-GFN | $86.81 \pm 0.24$ | $87.52 \pm 0.18$ | $81.80 \pm 0.59$ | $79.14 \pm 0.78$ |
| SubTB-GFN | $85.25 \pm 0.46$ | $86.67 \pm 0.57$ | $78.34 \pm 0.67$ | $74.10 \pm 0.10$ |
| MaxEnt RL (Soft DQN) | $80.27 \pm 0.52$ | $79.01 \pm 0.03$ | $73.52 \pm 0.72$ | $70.11 \pm 1.13$ |
| MaxEnt RL (M-DQN) | $89.55 \pm 0.51$ | $89.99 \pm 0.43$ | $78.28 \pm 0.98$ | $72.40 \pm 0.73$ |
| RPE (Ours) | $\mathbf{99.81 \pm 0.19}$ | $\mathbf{100.0 \pm 3.97}$ | $\mathbf{100.0 \pm 2.45}$ | $\mathbf{97.68 \pm 3.03}$ |

## D. Discussion

In this section, we discuss the limitations of RPE and the broader impact of our work. Although RPE obtains stronger performance compared with a thorough set of baselines in various benchmarks, there are several directions for future improvements. The current implementation computes $P_F(s'|s)$ through separate forward passes for each child state $s'$, which presents opportunities for computational optimization. Additionally, like many local credit assignment methods, RPE's effectiveness in environments with very long trajectories could be further improved, building upon recent advances

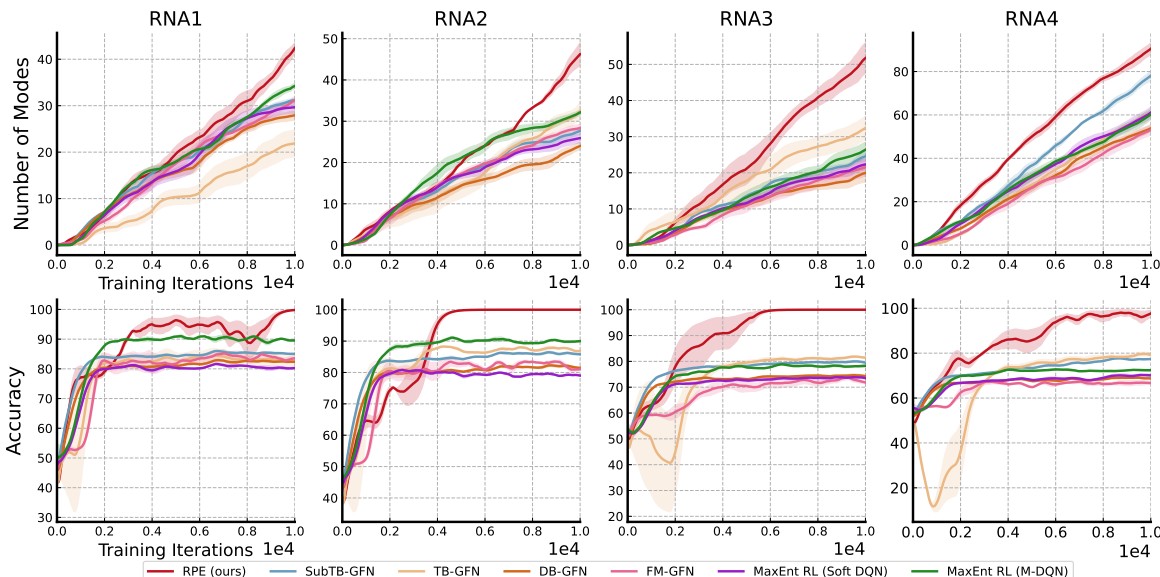

Figure 9: $P_B$ is fixed to be uniform for GFlowNets methods. *Top row:* number of modes discovered over training across 3 random seeds. *Bottom row:* Accuracy over training across 3 random seeds.

Table 2: The number of modes discovered in different RNA generation tasks.

|  | L14_RNA1 | L14_RNA2 | L14_RNA3 | L14_RNA4 |
|---|---|---|---|---|
| FM-GFN | $31 \pm 2$ | $28 \pm 2$ | $21 \pm 5$ | $53 \pm 5$ |
| DB-GFN | $32 \pm 4$ | $30 \pm 2$ | $25 \pm 4$ | $61 \pm 2$ |
| TB-GFN | $34 \pm 2$ | $31 \pm 2$ | $31 \pm 5$ | $79 \pm 7$ |
| SubTB-GFN | $32 \pm 3$ | $29 \pm 1$ | $27 \pm 4$ | $68 \pm 3$ |
| MaxEnt RL (Soft DQN) | $30 \pm 2$ | $26 \pm 2$ | $22 \pm 3$ | $61 \pm 5$ |
| MaxEnt RL (M-DQN) | $34 \pm 1$ | $32 \pm 1$ | $22 \pm 3$ | $60 \pm 5$ |
| RPE (ours) | $\mathbf{42 \pm 2}$ | $\mathbf{46 \pm 6}$ | $\mathbf{52 \pm 9}$ | $\mathbf{90 \pm 6}$ |

in temporal credit assignment (Malkin et al., 2022; Madan et al., 2023; Pan et al., 2023a). While specific scenarios (e.g., the HyperGrid task) may require additional consideration due to trajectory-dependent $g$-values, our work uncovers previously overlooked connections between GFlowNets and policy evaluation. This theoretical bridge not only deepens our understanding of both frameworks, but also opens up new research directions for future work to build upon, as demonstrated by our simplified yet effective training strategies.

**Differences between RPE and Soft Policy Iteration.** Our proposed RPE shares conceptual connections with Soft Policy Iteration (SPI), while establishing key distinctions that highlight the novelty of our work: (1) SPI necessitates iterative cycles of policy evaluation and policy improvement to derive the desired policy, while RPE evaluates only the uniform policy, transforming its value function to flow functions to achieve reward matching. (2) SPI and 'control as inference' framework both explicitly incorporate entropy regularization terms to ensure policy stochasticity, while RPE achieves superior reward-matching performance through standard policy evaluation utilizing a summarization operator, eliminating the need for explicit regularization. Our experimental results demonstrate RPE's superior performance compared to both SoftDQN and MunchausenDQN (which are advanced soft RL algorithms for discrete environments) in terms of the standard GFlowNets evaluation protocol.

**Assumptions in Theorem 4.2.** Theorems 4.1 & 4.2 are formulated within the context of GFlowNets, which address constructive tasks by sampling compositional objects through a sequence of constructive actions. These tasks are modeled as Directed Acyclic Graphs (DAGs), where agents do not revisit the same state within a single episode, ensuring the graph remains acyclic.

The path-invariance condition in Theorem 4.2 requires that the branching ratio $g(\tau, s_t)$ is identical for all trajectories reaching the same state $s_t$. This assumption is satisfied in a broad range of domains within the GFNs literature, beyond just tree-structured problems (e.g., sequence/language generation (Hu et al., 2024)), where the local branching structure exhibits symmetry properties inherent to several compositional generation tasks.

However, as we discuss in Section 4.1.2, there are certain cases, e.g., the HyperGrid task with boundary conditions (where task symmetry fails for states at edges), where this assumption does not hold. We view this limitation as a natural consequence of the simplicity of our method. Yet, it is precisely this simplicity that makes our findings both surprising and impactful, as they provide the first rigorous characterization of when GFNs and policy evaluation align.

