# OpenReview forum: "Random Policy Evaluation Uncovers Policies of Generative Flow Networks"
_ICML.cc/2025/Conference — ICML 2025 poster_

### Official Review · Reviewer_trFY · 2025-03-11

**Overall Recommendation:** 2

**Summary:**

This paper studies generative flow networks (GFlowNet) where one aims to learn a policy which generates states with probability proportional to the rewards of these states, contrast with the goal of reward maximization in typical reinforcement learning. This work tries to build the connection between value function and flow function in this framework. By transforming the reward, they can show an equality between a scaled value function and the flow function. With this finding, they interpret GFlowNet from the perspective of value iteration. What's more, they implemented experiments to show this connection.

**Claims And Evidence:**

Yes, the claims made in this paper are supported by the main theorems (Theorem 4.1 and Theorem 4.2).

**Essential References Not Discussed:**

No, I didn't see any essential related works that are not discussed.

**Experimental Designs Or Analyses:**

No, I didn't check the soundness of the experiments.

**Methods And Evaluation Criteria:**

It is not applicable to this paper. The focus of this paper is to establish the connection between two existing concept (or method), value function in value iteration and flow function in GFlowNet.

**Other Comments Or Suggestions:**

1. In Appendix A, the authors include an illustration to show the connection of value function and flow functions, but they only give a figure title, it would be better to at least elaborate it.
2. Missing proof for Theorem 4.1 in Appendix.

**Other Strengths And Weaknesses:**

Strength:
1. To some extent, this work shows potential results that policy evaluation can achieve beyond the evaluation itself or serving for policy improvement.

Weaknesses:
1. The theory developed in this paper provides very limited new insights compared with the existing results. For example, in Theorem 4.1 (the main theorem), they build the connection $V(s_t)=F(s_t)\Pi_{i=0}^{t-1}A(s_i)$ by transforming rewards. However, with this flow function, one can derive the forward policy $\pi(s'|s)=\frac{V(s')}{\sum_{s''}V(s'')}$ where $s''$ denotes all possible subsequent state from $s$. It is a known result already established in Proposition 1 in [1]. What's new here is just transforming the reward such that one can directly write  the flow function to the value function. It is not significant enough to serve as main contributions for a paper, let alone in GFlowNet, people care more about the forward policy, Theorem 4.1 didn't bring anything new about the forward policy.
2. In Theorem 4.2, the constraint that any trajectories $\tau_1$ and $\tau_2$ that visit $s_t$ should satisfy $g(\tau_1, s_t)=g(\tau_2,s_t)$ is too strong, which makes the interpretation only applicable to very limited graphs. The authors should dig deeper into it.

[1]. Bengio E, Jain M, Korablyov M, et al. Flow network based generative models for non-iterative diverse candidate generation[J]. Advances in Neural Information Processing Systems, 2021, 34: 27381-27394.

**Questions For Authors:**

My main concerns are listed in the weaknesses part, at the moment I don't have questions towards the understanding of this paper.

**Relation To Broader Scientific Literature:**

The key contribution in this work is to build the connection between the value function and flow function in GFlowNet. The most related result in previous works is Proposition 1 in [1]. This work provides more straightforward interpretation between value function and flow function beyond that result.

[1]. Bengio E, Jain M, Korablyov M, et al. Flow network based generative models for non-iterative diverse candidate generation[J]. Advances in Neural Information Processing Systems, 2021, 34: 27381-27394.

**Theoretical Claims:**

Yes, I checked the correctness of the main theorems in this work, i.e. Theorem 4.1 and 4.2. They seem correct to me.

---

> ### Author Rebuttal · Authors · 2025-03-30
>
> Thanks to the reviewer for the time and effort in the feedback. We would like to clarify the novelty and significance of our contributions as follows:
> >Q1: Concerns regarding insights: Theorem 4.1 didn't bring anything new about the forward policy.
>
> While the reward transformation is a key component in our algorithm, our work’s novelty and impact extend far beyond this step. The focus of our work is not to connect the 'value functions and flow functions in the GFlowNet', but to address a critical gap in the literature by bridging **GFlowNets** and **standard (non-MaxEnt) RL**, an underexplored and overlooked connection that holds broad implications for both theory and practice. Unlike previous works that necessitate flow-matching or entropy regularization [1,2,3], we demonstrate that a GFlowNet forward policy with reward-matching properties can emerge naturally from standard RL components like policy evaluation.
>
> The core of GFlowNets' research is not about defining forward policies in different formulations ($\pi(x) \propto R(x)$), but rather developing effective methods to learn these policies that sample proportionally to rewards. Proposition 1 in [1] explains why MaxEnt RL (Buesing et al., 2019 Haarnoja et al., 2017) are biased by the number of trajectories leading to each terminal state in non-injective environments (i.e., $\pi(x)=\frac{n(x)R(x)}{\sum_{s' \in X}n(x')R(x')}$). In contrast, our Theorems 4.1-4.2 provide distinctive contributions, serving as the foundation of RPE to demonstrate how a standard RL component (policy evaluation) can surprisingly achieve reward-matching properties without requiring specialized and sophisticated flow-matching objectives or entropy regularization. This fundamentally redefines how GFlowNets can be understood and implemented.
>
> - **Novel theoretical contributions**: We uncover an unexpected bridge between standard (non-MaxEnt) RL and GFlowNets, showing that a basic component in RL, i.e., policy evaluation, can naturally achieve the same reward-matching property as in GFlowNets with minimal modifications. **Our work is not a trivial reformulation or mere reward transformation, but offers a fundamentally new insight into the capabilities of standard RL components**. Our analysis demonstrates that GFlowNets’ core mechanisms can be reinterpreted through the lens of random policy evaluation in standard RL. This reshapes our understanding of GFlowNets’ learning dynamics, resolving an open question in the field.
> - **Algorithmic contributions**: Based on our main theoretical results in Theorems 4.1-4.2, we develop a simple yet effective RPE algorithm using only policy evaluation, which eliminates the need for complex flow-matching conditions [1,2,3]. Despite its simplicity, our extensive experiments show that RPE matches or exceeds the performance of well-established GFlowNet methods [1,2,3], validating both our theoretical insights and practical utility.
> - **Expanding understanding**: Our work provides novel insights into how reward-matching properties can naturally emerge from policy evaluation of a random policy under appropriate reward transformation. This expands the important understanding that was overlooked and opens up promising directions for future research at the intersection of RL and GFlowNets.
>
> In summary, our work makes substantial contributions by revealing a novel, simpler way to achieve reward-matching policies in GFlowNets through standard RL components under appropriate conditions.
>
> [1] Bengio et. al., Flow Network based Generative Models for Non-Iterative Diverse Candidate Generation, NeurIPS 2021
>
> [2] Bengio et. al., GFlowNet Foundations, JMLR 2023.
>
> [3] Malkin et. al., Trajectory balance: Improved credit assignment in GFlowNets, NeurIPS 2022
> >Q2: Concerns regarding the assumption in Theorem 4.2
>
> Thanks for your question. Please refer to our response to Reviewer ghFJ's Q1 for a detailed discussion due to space limitations.
> >Q3: Questions about Appendix A.
>
> Thanks for your question. We will elaborate on more details in Appendix A in the revision to explain the connections between the flow/value functions in GFlowNets, MaxEnt RL, and our RPE approach.
> >Q4: About proof for Theorem 4.1
>
> Thanks for your question. We clarified in Appendix B (lines 674-675) that Theorem 4.1 represents a special case of the more general Theorem 4.2. For the sake of conciseness and to avoid redundancy in the presentation, we chose to omit the explicit proof of Theorem 4.1 in the appendix (the proof can be readily derived by following the same logical framework presented in the proof of Theorem 4.2, with $B(s_{i+1}) = 1$ and there is exists exactly one path from $s_0$ to any state). To ensure completeness and facilitate better understanding, we will include the proof for this special case for Theorem 4.2 in the revision.
>
> We hope these clarifications effectively address your concerns and highlight our key contributions. We welcome any further questions or discussions.

---

> > ### Comment · Reviewer_trFY · 2025-04-04
> >
> > Thank the authors for their efforts on the rebuttal. After reading the elaborations, I am still having main concerns unsolved. There are usually two settings discussed for GFlowNet problem, one is bijective setting or tree MDP, another is non-injective setting or DAG. For tree MDP which is trivial, it is clearly stated in the existing work [1] that
> >
> > "Interestingly, in such a case one can express the pseudo-value of a state V(s) as the sum of all the rewards of the descendants of s."
> > It already implies most results in this work instead of explicitly stating it is the value function of the random policy.
> >
> > The most contents of [1] is trying to solve the non-injective setting which is more challenging. However, in this work, it seems to me that the challenges faced by the perspective of value iteration in this setting is simply framed as an assumption which I pointed out in my official review. I think the authors should deep into this challenge setting instead of adding restrictions.
> >
> > Overall, I think the current work can be a good starting point towards this direction instead of a work explored the problem thoroughly that ready to be published.
> >
> > [1]. Bengio E, Jain M, Korablyov M, et al. Flow network based generative models for non-iterative diverse candidate generation[J]. Advances in Neural Information Processing Systems, 2021, 34: 27381-27394.

---

> > > ### Author Response · Authors · 2025-04-04
> > >
> > > Dear Reviewer trFY, UFa1, ghFJ, and wvJJ,
> > >
> > > Thanks to Reviewer trFY for the time in the follow-up comments. While we appreciate the engagement, which provides us the opportunity to further clarify our work, **we identified several key points that we believe stem from misunderstandings or factual errors (especially after private correspondence with authors of [1]).** We address these concerns point-by-point to ensure our insights, contributions, and methods are correctly understood.
> > >
> > > **1. On Proposition 1 in [1] and its difference with RPE**
> > >
> > > > This work is "a known result already established in Proposition 1 in [1] -- implies most results in this work.
> > >
> > > We respectfully clarify that this is a misunderstanding of both Proposition 1's role in [1] and our contributions.
> > >
> > > - **Proposition 1 in [1] highlights biases introduced by MaxEnt RL in DAG structures ([2])**, particularly how value functions in MaxEnt RL (denoted by $\hat{V}$, which modified the RL objective with entropy regularization, that is different from the traditional value function $V$ in standard non-MaxEnt RL studied in our work) are biased by the number of paths to terminal states. In addition, **in our case, the value is NOT a sum of children values (and NOT in log-scale -- see [2] for full analysis), but mean instead (and also with other developments for a practical algorithm with new insights)**. While both MaxEnt and non-MaxEnt RL involve values, their definitions are different. Thus, Proposition [1] is distinct from our work, which theoretically and empirically demonstrates how policy evaluation for a random policy (non-MaxEnt RL) can yield value functions that effectively derive reward-matching policies.
> > > - More importantly, Proposition 1 is analytical in nature, and is not about providing a tractable solution for learning reward-matching policies. Instead, the **Flow Matching** algorithm introduced in [1] builds on this analysis to offer a tractable method for policy learning (while our RPE provides a new and different method).
> > >
> > > In contrast, our work offers a fundamentally different and tractable approach. We propose a **policy evaluation variant for a FIXED RANDOM policy**, grounded in our theoretical finding that the random policy's value function (with scaled rewards) naturally aligns with the flow function in GFlowNets. `This insight provides a new and simpler alternative to the FM objective in [1] and other more complex developed objectives, including DB and TB, achieving competitive performance on standard and widely-adopted benchmarks.`  Please refer to our response to Q1 for Reviewer trFY for more details of our distinctive contributions (due to space limitations).
> > >
> > > **2. Mischaracterization of our method as "Value Iteration"**
> > >
> > > > ... by the perspective of value iteration in this setting ....
> > >
> > > > ... they interpret GFlowNet from the perspective of value iteration
> > >
> > > > ... the connection between ... value function in value iteration and flow function in GFlowNet.
> > >
> > > These statements mischaracterize our method, which is based on **policy evaluation**, not value iteration. Policy evaluation is a foundational component of policy iteration, but it does not involve policy improvement (a key aspect of value iteration) [3].
> > >
> > > This distinction is critical since our work simplifies GFlowNet training by reframing it as policy evaluation, avoiding the complexity of other complex learning objectives while achieving competitive performance across benchmarks.
> > >
> > > **3. Clarification on the role of the assumption**
> > >
> > > > The challenges faced by the perspective of value iteration in this setting is simply framed as an assumption.
> > >
> > > As thoroughly explained in our rebuttal, we emphasize that the assumption we make is satisfied across a broad range of domains within the GFlowNets literature (extensive tree and non-tree scenarios, `as elaborated in our response to Q1 for reviewer ghFJ`).
> > >
> > > Overall, we appreciate the Reviewer’s time and effort in reviewing our work. **While we acknowledge the value of critical feedback, we hope the above clarifications address the factual errors and misunderstandings raised.**
> > >
> > > **We believe our work provides a novel and meaningful contribution to GFlowNets research, which is also recognized by other reviewers both in terms of theoretical insights and practical performance:** Our work establishs a "`novel connection`" (UFa1, wvJJ) between GFlowNets and a fundamental RL problem, supported by "`well-motivated`" (ghFJ) and "`sound theoretical claims`" (ghFJ), with "`promising`" (UFa1) and "`SOTA`" (wvJJ) empirical results achieved by "`novel`" (wvJJ) RPE method.
> > >
> > > **References**
> > >
> > > [1] Bengio, Emmanuel, et al. Flow network based generative models for non-iterative diverse candidate generation. NeurIPS 2021.
> > >
> > > [2] Buesing et al. Approximate inference in discrete distributions with monte carlo tree search and value functions. AISTATS 2020.
> > >
> > > [3] Sutton et al. Reinforcement learning: An introduction. Cambridge: MIT press, 1998.

---

### Official Review · Reviewer_wvJJ · 2025-03-12

**Overall Recommendation:** 4

**Summary:**

The authors present a relationship between Generative Flow networks and RL via policy evaluation. Specifically, the authors claim that the value function obtained from evaluating a uniform policy is closely related to the flow function in GFlowNets.

**Claims And Evidence:**

Yes

**Essential References Not Discussed:**

n/a

**Experimental Designs Or Analyses:**

Yes

**Methods And Evaluation Criteria:**

Yes

**Other Comments Or Suggestions:**

n/a

**Other Strengths And Weaknesses:**

**Strengths**
- The authors’ insight that evaluating a random policy approximately matches the flow function leads to a simple, practical, and novel algorithm that bridges RL and GFlowNets.
- The proposed RPE algorithm empirically validates the claim that random policy evaluation matches the flow function with high accuracy results on various benchmarks.
- The algorithm achieves SOTA results on discovering diverse solutions while maintaining high accuracy when compared to both GFlowNets and max-entropy RL.

**Weaknesses**
- The Max Entropy RL baselines, specifically DQN, are not state of the art. Why was the proposed method not compared against a more recent method like Soft Actor-Critic (SAC)?

**Questions For Authors:**

n/a

**Relation To Broader Scientific Literature:**

This paper broadly relates to solving sequential decision making problems with reinforcement learning and/or generative flow networks. Here the authors specifically investigate benchmarks where the solution space is multimodal i.e. there are many diverse solutions that achieve high reward since generative flow networks are well equipped to capture the multiple modalities.

**Theoretical Claims:**

No

---

> ### Author Rebuttal · Authors · 2025-03-30
>
> We thank the reviewer for the useful feedback and positive assessment of our work, and for noting that our work builds a simple, practical, and novel algorithm that bridges RL and GFlowNets. We carefully address your concerns as follows:
> >Q1: Questions about Max Entropy RL baselines
>
> Thanks for your insightful question! Our choice of Soft DQN and Munchausen DQN as the max-entropy RL baselines was guided by prior research [1], considering the discrete action spaces in our experimental settings. While SAC was originally designed for continuous control problems and would require discretization in our context, Soft DQN is naturally suited for discrete spaces, with Munchausen DQN representing an enhanced variant with documented performance advantages.
>
> To directly address the reviewer's concern about SAC, we conduct additional experiments incorporating SAC as a baseline on the RNA1 generation task, following the cleanRL's implementation. The results illustrated in Figure 2 in https://anonymous.4open.science/r/RPE-added-experiments-57A9 show that SAC achieves comparable performance to Soft DQN in terms of mode discovery but is less efficient in terms of accuracy, which is consistent to prior research [1] demonstrating that discrete SAC typically exhibits inferior performance compared to Soft DQN and Munchausen DQN in discrete action spaces for similar tasks. **Notably, our proposed RPE method achieves superior performance across all metrics against these established baselines.**
>
> [1] Tiapkin et. al., Generative Flow Networks as Entropy-Regularized RL, AISTATS 2024

---

> > ### Comment · Reviewer_wvJJ · 2025-04-04
> >
> > I appreciate the authors providing an additional discrete-SAC baseline and I am satisfied with their response. Several other reviewers have brought up concerns about the correctness of theorems, whether the assumptions made in the paper hold for the non-tree DAG case, etc which are valid. Another reviewer had issues with the novelty of this method. While I am not entirely familiar with the literature on GFlowNets, it seems this connection between the value function under uniform policy evaluation and GFlowNets in the non max-Ent RL setting has not been discovered or previously studied. If that's the case, then I believe the simplicity of this approach and the connection it reveals will be beneficial to both RL and GFlowNet communities and thus I will maintain my score.

---

> > > ### Author Response · Authors · 2025-04-05
> > >
> > > Dear reviewer wvJJ,
> > >
> > > Thanks for your time and effort in the follow-up comment! We are pleased to know that our rebuttal has addressed your concerns, and we appreciate the supportive feedback and for recognizing the simplicity and potential impact of our work!
> > >
> > > We would like to take this opportunity to summarize the novelty and contributions of our work:
> > > - Theoretical connections and algorithm design: We investigate and generalize the theoretical connections between value functions (in standard non-MaxEnt RL) and flow functions in both tree-structured and non-tree-structured DAGs. While prior work has explored certain theoretical connections for values in MaxEnt RL and flows in GFNs and in restricted tree-structured cases [1,2], our work significantly extends these ideas to the more general and widely applicable non-tree DAG setting, which provides new theoretical insights to account for the structural complexity of DAGs. (Additionally, as we have thoroughly explained in the rebuttal (e.g., response to Q1 for reviewer ghFJ and further clarification for reviewer UFa1), we would like to remark that the assumption we make is satisfied in a broad range of domains within the GFlowNets literature, including both tree and DAG tasks.) Building on these theoretical insights, we propose a novel and practical algorithm, RPE, leveraging our established theoretical connection between value functions and flows, which has not been previously discovered or studied. RPE is built upon ideas from policy evaluation in RL and forward policy structure in GFlowNets to offer a novel and unified approach that is both theoretically grounded and empirically validated.
> > > - Empirical validation across benchmarks: Our method, RPE, achieves superior performance compared to existing GFlowNet algorithms and MaxEnt RL methods across diverse benchmarks (including both tree and DAG scenarios). This can be attributed to a fundamental algorithmic design advantage of RPE: it evaluates a fixed uniform policy $\pi$, while both standard GFlowNets and MaxEnt RL algorithms need to estimate flows/values for continuously evolving policies. This distinctive approach helps RPE eliminate key sources of non-stationarity that can contribute to training instability. In addition, RPE adopts a simplified parameterization that learns only the flow function $F_{\theta}$, from which the sampling policy can be directly derived, which can reduce the potential approximation error from function approximators [3].
> > >
> > > While there have been a number of works studying the connection between GFlowNets and variational inference [4], MCMC [5], generative models [6], and MaxEnt RL (which modified RL objectives) [1], our work provides distinctive and novel contributions by bridging the gap between standard non-MaxEnt RL and GFlowNets in general tree-structured and a number of non-tree-structured DAG problems. We believe our work provides significant new insights and practical tools for both communities. Furthermore, our work is not limited to theoretical insights. By extending the framework to non-tree DAGs, designing the RPE algorithm, and demonstrating its effectiveness through extensive experiments, we believe our contributions address both theoretical and practical gaps in the field, offering a new algorithm that has not been studied in previous works.
> > >
> > > Overall, we sincerely thank you for your time and effort in reviewing our work, as well as for your valuable suggestions and feedback!
> > >
> > > [1] Tiapkin et. al., GFlowNets as Entropy-Regularized RL, AISTATS 2024
> > >
> > > [2] Yoshua Bengio et. al., GFlowNet Foundations, JMLR 2023.
> > >
> > > [3] Shen et. al., Towards Understanding and Improving GFlowNet Training, ICML 2023
> > >
> > > [4] Malkin et. al., GFlowNets and variational inference, ICLR 2023
> > >
> > > [5] Tristan et. al., Generative flow networks: a markov chain perspective, arXiv 2023
> > >
> > > [6] Zhang et. al., Unifying generative models with GFlowNets and beyond, ICML Workshop 2022

---

### Official Review · Reviewer_ghFJ · 2025-03-14

**Overall Recommendation:** 3

**Summary:**

The paper presents a connection between GFlowNets and non max-ent RL. It leverages insights in the special case with a uniform policy to establish the connection, which leads to the development of the RPE algorithm. Empirical results suggest that RPE achieves competitive performance with existing GFlowNet training and entropy-regularized RL baselines.

### Update after rebuttal

I thank the authors for their extensive efforts in the rebuttal. The additional clarifications on the applicability of scaling factors and additional evidence on the empirical advantage of the proposed method well addressed my concerns.

While I'm not familiar with the GFlowNet literature and only have an understanding of MaxEnt RL, revealing the connection between value functions and flow functions in non-DAG scenarios seems novel and serves as an important step to bridge the research efforts along these two topics. The assumptions in the framework seem to be lenient enough to cover a wide range of environments in the literature.

For the reasons above, I kept my score to be positive.

**Claims And Evidence:**

Claims are well-supported.

**Essential References Not Discussed:**

N/A

**Experimental Designs Or Analyses:**

Empirical validations involve multiple standard benchmark environments. However, it's not shown how the asymptotic performance of the proposed algorithm compares to baselines, for RNA1-RNA4 (Fig. 5) beyond training iterations 1e4.

**Methods And Evaluation Criteria:**

* The approach is well-motivated, but it is unclear if RPE retains good performance in environments with non-uniform backward policies.

**Other Comments Or Suggestions:**

N/A

**Other Strengths And Weaknesses:**

Weaknesses:
* In the non-tree DAG case, requiring the scaling factor to be constant across paths to any state is a strong assumption that holds in the benchmark being considered but not necessarily beyond. This is discussed in Section 4.2 and does not decrease the value of the theoretical insights, but limits the applicability of the proposed algorithm in practice.

**Questions For Authors:**

* How does the algorithm compare with baselines in terms of training variance and stability?

**Relation To Broader Scientific Literature:**

Relation to the literature is well explained in the paper, by introducing a few works linking GFlowNets to MaxEnt RL and highlights the contribution of connecting GFlowNets with standard RL in this work.

**Theoretical Claims:**

* Theoretical claims are sound.

---

> ### Author Rebuttal · Authors · 2025-03-30
>
> Thanks for your valuable feedback and for a positive assessment of our work! We carefully address your concerns as follows:
>
> >Q1: Concerns about path invariance scaling factor $g$
>
> Thanks for your question. We would like to emphasize that this condition is satisfied in a broad range of domains of GFlowNets literature, including most standard benchmarks and important application areas: 1) all tree-structured problems naturally satisfy this condition, e.g., sequence/language generation [1,3], red-teaming [2], and prompt optimization [4]; 2) non-tree structured DAGs where the number of parent and children states remains independent of the specific trajectory to the state, like the widely-studied set generation tasks[4], and real-world applications including feature selection[5], recommender systems[6], experimental design optimization[1], portfolio construction, etc. The key requirement is that the task can be modeled as a DAG where the ratio of children states or parent states remains consistent regardless of the specific path taken to reach the state. These examples demonstrate that the condition applies to a wide range of practical generative tasks within the GFlowNet framework.
>
> However, as noted in Section 4.1.2, we acknowledge that RPE's simplicity imposes limitations in certain environments, such as the toy HyperGrid task with boundary conditions. Yet, this simplicity is precisely what makes our findings both surprising and significant. Our work provides the first rigorous characterization of when GFlowNets and policy evaluation align, bridging a critical gap in understanding the relationship between (non-MaxEnt) RL and GFlowNets. This fundamental insight unlocks new possibilities and advances both fields by leveraging their complementary strengths.
>
> Despite being based on basic policy evaluation principles for a uniform policy, our method achieves performance comparable to many well-developed GFlowNet algorithms while maintaining remarkable simplicity across diverse environments, making it a valuable foundation for future exploration in GFlowNets and RL research.
> >Q2: Questions about training variance and stability
>
> Thanks for your question. As demonstrated in Figures 3,4,5,6 in the main text, RPE exhibits both consistently improving performance and remarkably stable learning throughout the training process. It is noteworthy that RPE has a fundamental algorithmic design difference (advantage): it evaluates a fixed uniform policy $\pi$, while both standard GFlowNets and MaxEnt RL algorithms need to estimate flows/values for continuously evolving policies. This distinctive approach helps RPE eliminate key sources of non-stationarity that can contribute to training instability.
>
> To ensure statistical reliability, all experimental results are averaged across multiple random seeds following evaluation standards in previous works [7]. For the RNA1 generation task, `RPE demonstrates the lowest standard deviation among all baseline methods` (please see Table 1 in https://anonymous.4open.science/r/RPE-added-experiments-57A9). The reduced variance of RPE can be attributed to reduced complexity in the learning objective through fixed policy evaluation and simplified parameterization that learns only the flow function $F_{\theta}$, different from privious state-of-the-art GFlowNets method like trajectory balance (TB) [7], which can incur high variance due to the reliance on Monte-Carlo estimation techniques.
>
> >Q3: Performance of RPE beyond training iterations 1e4.
>
> To further evaluate RPE's sustainability beyond the standard training regime, we extend our experimental analysis by training for 1e5 iterations (10$\times$) on the RNA1 task, and compare it against the most competitive algorithms from both Soft RL and GFlowNets paradigms, including Munchausen-DQN and TB in this task. The results presented in Fig.1 in https://anonymous.4open.science/r/RPE-added-experiments-57A9 demonstrate that RPE maintains its performance advantage throughout the extended training period, which consistently outperforms these top-performing baselines, validating its effectiveness as a stable and high-performing approach.
>
> **References**
>
> [1] Jain et. al., GFlowNets for AI-Driven Scientific Discovery, Digital Discovery, 2023
>
> [2] Lee et. al., Learning diverse attacks on large language models for robust red-teaming and safety tuning, ICLR 2025
>
> [3] Hu et. al., Amortizing intractable inference in large language models, ICLR 2024
>
> [4] Yun et. al.,Learning to Sample Effective and Diverse Prompts for Text-to-Image Generation, CVPR 2025
>
> [4] Pan et. al., Better Training of GFlowNets with Local Credit and Incomplete Trajectories, ICML 2023
>
> [5] Ren et. al., ID-RDRL: a deep reinforcement learning-based feature selection intrusion detection model, Scientific Reports 2022
>
> [6] Liu et. al., Generative Flow Network for Listwise Recommendation, KDD 2023
>
> [7] Malkin et. al.,Trajectory balance: Improved credit assignment in GFlowNets, NeurIPS 2022

---

### Official Review · Reviewer_UFa1 · 2025-03-15

**Overall Recommendation:** 2

**Summary:**

This paper explores a connection between GFlowNets and policy evaluation in the specific setting of undiscounted Reinforcement Learning with only terminal rewards. The central idea revolves around the flow constraint in GFlowNets, which shows that the flow out of a state must equal the total in-flow from its successor states. The authors draw a parallel between this constraint and the objective of policy evaluation, where the value of a state is the average of the values of its descendant states. The key difference identified is a normalization factor (the number of descendants) in policy evaluation. The paper proposes scaling the flow function in GFlowNets to account for this normalization, suggesting a potential way to perform policy evaluation using GFlowNets, which is shown to work for tree-structured state spaces.

## update after rebuttal

After discussing with the author, I will maintain my original assessment given the limitations of the methods and their similarity to existing works

**Claims And Evidence:**

The paper claims a connection between GFlowNets and policy evaluation and proposes a method to bridge this gap, particularly for undiscounted RL with only terminal rewards. The theoretical result for the tree case (Theorem 4.1) seems plausible based on the explanation. However, I'm not sure about if it's correct for the general DAG case (Theorem 4.2).

**Essential References Not Discussed:**

However, a more comprehensive understanding of the GFlowNet and policy evaluation literature might reveal missing relevant works.

**Experimental Designs Or Analyses:**

The experiments are conducted on predicting DNA or RNA sequences andn molecule generation. I wonder why the comparison on a standard RF setting is not presented.

**Methods And Evaluation Criteria:**

The proposed method involves scaling the flow function F(s) in GFlowNets based on the number of "branches" or cumulative inverse of the number of descendants to align with the policy evaluation objective. The evaluation seems to primarily focus on the theoretical derivations, with limited mention of empirical evaluation or specific evaluation criteria. I'm not sure about the motivation behind this specific approach and its advantages, as well as its relation to existing policy evaluation methods like Soft Policy Iteration.

**Other Comments Or Suggestions:**

- The authors should rigorously define all variables in the equations to improve clarity.
- The ambiguity surrounding the definition of R′(x)=R(x)g(τ,x) in the context of DAGs with multiple trajectories reaching the same terminal state needs to be addressed. How is the τ-dependence resolved?
- The condition mentioned in lines 629-630, "If any trajectories τ_1 and τ_2 that visits s_t satisfy g(τ_1,s_t)=g(τ_2,s_t)," needs to be clearly stated whether it is an assumption and, if so, how it can be generally satisfied in a DAG. In line 068 (2nd column), the backward probability P_B(s∣s′) seems to have a typo and should likely be P_B(s′∣s).
- The paper would benefit from a clearer explanation of the motivation behind using this specific approach for policy evaluation. What are the potential advantages or insights it offers compared to existing methods?
- A discussion on how this work relates to Soft Policy Iteration and the "control as inference" framework would help contextualize the contribution.

**Other Strengths And Weaknesses:**

Strengths: The paper attempts to establish a novel connection between GFlowNets and a fundamental problem in Reinforcement Learning, policy evaluation. The result for the tree case (Theorem 4.1) seems promising.

Weaknesses: The main weakness lies in the ambiguity and potential flaw in the extension to the general DAG case (Theorem 4.2), particularly concerning the τ-dependence in the definition of R′(x). The lack of clarity in definitions and assumptions, such as the condition in line 629-630, also detracts from the paper's strength. Furthermore, the motivation for this specific approach and its relation to existing RL methods are not clearly articulated. The paper also lacks details on experimental validation.

**Questions For Authors:**

1. In the definition of R′(x)=R(x)g(τ,x) for the general DAG case (Theorem 4.2), how is the τ-dependence handled given that multiple trajectories can reach the same terminal state x? Please clarify why R′(x) on the left-hand side would be τ-independent.
2. Regarding the condition stated in lines 629-630: "If any trajectories τ_1 and τ_2 that visits s_t satisfy g(τ_1,s_t)=g(τ_2,s_t)," is this an assumption that needs to hold for your theoretical results to be valid? If so, under what conditions on the DAG structure and the transition probabilities would this assumption be satisfied in general?
3. In line 068 (2nd column), for the backward probability, is P_B(s∣s′)=F(s→s′)/F(s′) a typo? Should it be P_B(s′∣s)=F(s→s′)/F(s) or P_B(s′∣s)=F(s→s′)/F(s′)?
4. What is the specific motivation for using this approach to connect GFlowNets and policy evaluation? What advantages does it offer over existing policy evaluation techniques, particularly in the context of undiscounted RL with only terminal rewards?
5. How does this work relate to the literature on Soft Policy Iteration and the "control as inference" paradigm, where policies are often treated as uniform distributions? What are the key differences and similarities?

**Relation To Broader Scientific Literature:**

This work is related to Soft Policy Iteration (control as inference), where the policy is indeed often treated as uniform. The paper should discuss how this approach compares to and differs from existing methods in the context of policy evaluation and control.

**Theoretical Claims:**

I'm unsure about the validity of Theorem 4.2, which extends the connection to the general DAG case. The definition of g(τ,s_t) is clear. However, the subsequent definition of R′(x)=R(x)g(τ,x) is ambiguous. In a DAG, multiple trajectories τ can lead to the same terminal state x. This raises a concern as the left side of the equation (R′(x)) should ideally be independent of the specific trajectory taken, while the right side (R(x)g(τ,x)) explicitly depends on τ through g(τ,x). This suggests a potential flaw in the formulation for the general DAG case. The correctness of the proof for Theorem 4.2 is therefore in question.

---

> ### Author Rebuttal · Authors · 2025-03-30
>
> Thanks for your valuable suggestions and feedback! We carefully address your concerns as follows:
> >Q1: $\tau$-dependence regarding transformed rewards $R(x)g(\tau,x)$
>
> Thanks for your question! This concern is addressed by the assumption in Theorem 4.2: "For any trajectories $\tau_1$ and $\tau_2$ that visit $s_t$, they must satisfy the condition $g(\tau_1, s_t) = g(\tau_2, s_t)$." Under this assumption, the transformation from $R(x)$ to $R'(x)=R(x)g(\tau,x)$ is well-defined, since the transformed reward function $R'(x)$ remains consistent regardless of the specific trajectory taken to reach the terminal state $x$ (and this property naturally holds in a number of standard GFlowNet benchmarks -- Q2). We will also update the notation to eliminate any ambiguity.
> >Q2: questions about the condition g(τ_1,s_t)=g(τ_2,s_t)
>
> Thanks for your question. Indeed, this condition is a necessary assumption for the validity of Theorem 4.2.
>
> We remark that this assumption can be satisfied in a number of standard GFlowNets problems. Please refer to our response to Reviewer ghFJ's Q1 for a detailed discussion.
> >Q3: Definition of P_B
>
> Thanks for the question. This is not a typo, and $P_B$ is defined as $P_B(s|s')=\frac{F(s\to s')}{F(s')}$ following the GFlowNets literature (Eq. (17) in [1]), representing the probability of having come from state $s$ given that we are currently at state $s'$ (defined in terms of the flow).
> >Q4: Concerns regarding motivations and contributions.
>
> The motivation for connecting GFlowNets and policy evaluation stems from a critical gap in the current understanding of GFlowNets' relationship with RL frameworks. While previous research has extensively explored connections between GFlowNets and various ML methods, their link to RL has largely been limited to MaxEnt RL [2,3] via modified objectives with entropy regularization. However, the connection between GFlowNets and standard (non-MaxEnt) RL remains largely unexplored, despite the shared foundation of sequential decision-making. This gap limits cross-disciplinary insights (e.g., leveraging RL’s efficiency for GFlowNets or enriching RL with GFlowNets’ diversity).
>
> Our work fills this crucial gap and makes the following key contributions: (1) Surprisingly, we find that the flow functions in GFlowNets naturally correspond to value functions in uniform policy evaluation. This insight leads to our key finding that reward-matching effects, previously achieved through complex GFlowNets algorithms [1,4], **can be accomplished through simple policy evaluation.** (2) Based on this theoretical foundation, our proposed RPE method achieves competitive performance with reduced complexity. (3) Our work challenges the prevailing belief that GFlowNets are intrinsically tied to MaxEnt RL and provides a fundamentally new perspective on GFlowNets learning dynamics.
>
> In addition, traditional policy evaluation is merely a component for policy improvement; RPE reframes it to generate diverse, high-quality candidates via uniform policy evaluation with transformed rewards, expanding its utility to domains requiring both quality and diversity.
> >Q5: relations to Soft Policy Iteration (SPI) and the "control as inference" framework
>
> Thanks for your insightful question. Our work shares conceptual connections with SPI, while establishing key distinctions that highlight the novelty of our work: (1) SPI necessitates iterative cycles of policy evaluation and policy improvement to derive the desired policy, while RPE evaluates only the uniform policy, transforming its value function to flow functions to achieve reward matching. (2) SPI and 'control as inference' framework both explicitly incorporate entropy regularization terms to ensure policy stochasticity, while RPE achieves superior reward-matching performance through standard policy evaluation utilizing a summarization operator, eliminating the need for explicit regularization. Our experimental results demonstrate RPE's superior performance compared to both SoftDQN and MunchausenDQN (which are advanced soft RL algorithms for discrete environments) in terms of the standard GFlowNets evaluation protocol. We will add detailed discussions between them in the revision.
>
> >Q6: Comparison on a standard RF setting
>
> Since the goal of our work is to validate that a simple RPE algorithm can achieve competitive reward-matching performance as existing complex GFlowNets, we evaluate RPE and extensive baselines using well-established benchmarks in the GFlowNets literature [8,9], e.g., TFBind/RNA/molecule generation, which requires a critical balance between diversity and quality.
>
> [1] Yoshua Bengio et. al., GFlowNet Foundations, JMLR 2023.
>
> [2] Tiapkin et. al., Generative Flow Networks as Entropy-Regularized RL, AISTATS 2024
>
> [3] Tristan et al., Discrete probabilistic inference as control in multi-path environments, UAI 2024
>
> [4] Bengio et. al., Flow Network based Generative Models for Non-Iterative Diverse Candidate Generation, NeurIPS 2021

---

> > ### Comment · Reviewer_UFa1 · 2025-04-05
> >
> > Thank you for your response.
> >
> > Firstly, there's indeed an typo in the original paper. On page 2, in the second column, line 14, the definition of P_B(s|s′) is given as F(s → s′)/F(s). And the correct definition should be P_B(s|s′) = F(s → s′)/F(s'). I apologize, as I also made a similar error in my initial review, which may have caused the misunderstanding.
> >
> > Secondly, regarding Theorem 4.2, as we discussed, the current assumption might be quite restrictive, potentially limiting the theorem's applicability to a broader range of scenarios. It'll be better if this could be shown to hold under more general conditions (other than applying to only the tree stucture). Otherwise, I will likely maintain my current assessment.

---

> > > ### Author Response · Authors · 2025-04-05
> > >
> > > Dear reviewer UFa1,
> > >
> > > Thanks for your time and effort in the follow-up comment, and for pointing out the typo, which we will update in the revision. We also appreciate the opportunity to further clarify the generality of the assumption introduced in Theorem 4.2.
> > >
> > > We apologize that, due to space limitations, we could not elaborate on the generality of the assumption in the initial rebuttal and had to `defer our response to Reviewer ghFJ's Q1`. In this response, we aim to provide a detailed clarification of the reasoning behind this assumption, its applicability within the GFlowNets framework, and its implications for practical applications.
> > >
> > > Theorems 4.1-4.2 are formulated within the context of GFlowNets, which address **constructive tasks** by sampling compositional objects through a sequence of constructive actions. These tasks are modeled as **Directed Acyclic Graphs (DAGs)**, where agents do not revisit the same state within a single episode, ensuring the graph remains acyclic. This structural property of DAGs inherently supports the assumption in Theorem 4.2, **which can be satisfied in a broad range of domains within the GFNs literature, beyond just tree-structured problems** (e.g., sequence/language generation [1,2], red-teaming [3], and prompt optimization [4]):
> > > - **Non-tree structured DAGs:** `Many standard GFlowNets benchmarks satisfy this assumption` due to the constructive nature of these DAG tasks, where states are compositional objects built via sequences of actions. Specifically, in most practical scenarios,  the number of actions at a state $s_i$ ($|A(s_i)|$) and the in-degree of its child states ($|B(s_{i+1})|$) are inherently linked by the symmetry of compositional generation. For example, in tasks where objects are built via k-step additive processes (e.g., sets or sequences generation [1,5]), the number of valid actions at step $i$ (e.g., adding one of $n-i$ remaining elements) scales inversely with the in-degree of the resulting state (e.g., $i+1$ ways to remove an element to return to its parent). This relationship ensures the ratio $\frac{|A(s_i)|}{|B(s_{i+1})|}=\frac{n-i}{i+1}$, a trajectory-independent constant determined solely by the combinatorial rules of the task. Even in non-uniform DAGs (e.g., molecular graphs with variable branching like the prepend/append MDP studied in our paper), consistency arises when transitions satisfy local balance: if every path to $s_{i+1}$ requires a fixed number of parent states proportional to the action space at $s_i$, the ratio remains stable. Therefore, our assumption can naturally hold in these DAG tasks, since the ratios remain consistent regardless of the trajectory $\tau$. We remark that many practical examples with DAG structure also satisfy this assumption, including widely-studied set generation tasks [5], feature selection [6], recommender systems [7], and experimental design optimization. **Moreover, our empirical results validate RPE's superior performance on commonly studied DAG tasks like TFBind/RNA/Molecule sequence generation.** Thus, we argue that this assumption is not a restriction for GFlowNets problems, **since it broadly applies across extensive tree and non-tree (DAG) scenarios**.
> > >
> > > However, as we acknowledge in Section 4.1.2, there are certain corner cases, such as the toy HyperGrid task with boundary conditions (where task symmetry fails for states at edges), where this assumption may not hold. We view this limitation as a natural consequence of the simplicity of our method. **Yet, it is precisely this simplicity that makes our findings both surprising and impactful**, as our work provides the first rigorous characterization of when GFNs and policy evaluation align.
> > >
> > > Despite its simplicity, our method achieves performance comparable to existing GFlowNet algorithms across diverse benchmarks (including tree and DAG), offering a practical and theoretically grounded approach that bridges the gap between standard (non-MaxEnt) RL and GFNs. We believe this makes our work a valuable foundation for further exploration in both fields.
> > >
> > > We hope this response addresses the concern and demonstrates the generality and significance of our findings, and we will further expand the discussion of this assumption in the revision.
> > >
> > > [1] Jain et. al., GFlowNets for AI-Driven Scientific Discovery, Digital Discovery 2023
> > >
> > > [2] Hu et. al., Amortizing intractable inference in large language models, ICLR 2024
> > >
> > > [3] Lee et. al., Learning diverse attacks on large language models for robust red-teaming and safety tuning, ICLR 2025
> > >
> > > [4] Yun et. al.,Learning to Sample Effective and Diverse Prompts for Text-to-Image Generation, CVPR 2025
> > >
> > > [5] Pan et. al., Better Training of GFlowNets with Local Credit and Incomplete Trajectories, ICML 2023
> > >
> > > [6] Ren et. al., ID-RDRL: a deep reinforcement learning-based feature selection intrusion detection model, Scientific Reports 2022
> > >
> > > [7] Liu et. al., Generative Flow Network for Listwise Recommendation, KDD 2023

---

### Decision · Program_Chairs · 2025-05-01

**Decision:**

Accept (poster)

**Comment:**

This paper brings theoretical and algorithmic contributions. Notably, it establishes a connection between the value function of a random policy and the flow function of a GFlowNet which is novel. From this, they propose a new algorithm that they test on non-standard RL benchmarks (DNA and RNA sequence generation)

The strength of the paper is to create bridges between standard RL and GFlowNets which also makes it uneasy to review as knowledge in both fields are needed. Yet, the reviewers appreciated that the paper convincingly creates connections between the fields. The experiments seem to support the claim that the algorithmic contribution is real and outperforms other methods on the benchmarks.

Yet, some reviewers have raised concerns about the validity of hypotheses beyond special cases. The discussion didn't completely resolve these issues and the authors should take this into account in a revised version of the paper to adapt the claims.